# Potent and selective covalent inhibition of the papain-like protease from SARS-CoV-2

Brian C. Sanders [1,25] ✉, Suman Pokhrel [2,3,25], Audrey D. Labbe[1], Irimpan I. Mathews [4], Connor J. Cooper[1], Russell B. Davidson[1], Gwyndalyn Phillips [5], Kevin L. Weiss [5], Qiu Zhang[5], Hugh O'Neill [5], Manat Kaur[6], Jurgen G. Schmidt [7], Walter Reichard[8], Surekha Surendranathan[9], Jyothi Parvathareddy[9], Lexi Phillips[10], Christopher Rainville[11], David E. Sterner[11], Desigan Kumaran[12], Babak Andi [13], Gyorgy Babnigg [14,15], Nigel W. Moriarty [16], Paul D. Adams [16,17], Andrzej Joachimiak [14,18,19], Brett L. Hurst [10], Suresh Kumar [11], Tauseef R. Butt[11], Colleen B. Jonsson [8], Lori Ferrins [20], Soichi Wakatsuki [2,4,6] ✉, Stephanie Galanie [1,23], Martha S. Head[21,22,24] & Jerry M. Parks [1] ✉

Direct-acting antivirals are needed to combat coronavirus disease 2019 (COVID-19), which is caused by severe acute respiratory syndrome-coronavirus-2 (SARS-CoV-2). The papain-like protease (PLpro) domain of Nsp3 from SARS-CoV-2 is essential for viral replication. In addition, PLpro dysregulates the host immune response by cleaving ubiquitin and interferon-stimulated gene 15 protein from host proteins. As a result, PLpro is a promising target for inhibition by small-molecule therapeutics. Here we design a series of covalent inhibitors by introducing a peptidomimetic linker and reactive electrophile onto analogs of the noncovalent PLpro inhibitor GRL0617. The most potent compound inhibits PLpro with $k_{inact}/K_I = 9{,}600$ M$^{-1}$ s$^{-1}$, achieves sub-µM EC$_{50}$ values against three SARS-CoV-2 variants in mammalian cell lines, and does not inhibit a panel of human deubiquitinases (DUBs) at >30 µM concentrations of inhibitor. An X-ray co-crystal structure of the compound bound to PLpro validates our design strategy and establishes the molecular basis for covalent inhibition and selectivity against structurally similar human DUBs. These findings present an opportunity for further development of covalent PLpro inhibitors.

COVID-19 emerged globally with the rapid spread of the previously unrecognized beta-coronavirus SARS-CoV-2[1,2]. The virus is highly transmissible and can lead to severe, and in many cases life-threatening, respiratory disease. Few effective drugs have been developed to date, with molnupiravir[3] and nirmatrelvir[4] being the only currently available oral antivirals for treating SARS-CoV-2 infections. Although vaccines and therapeutic antibodies are effective in preventing COVID-19 or reducing its severity, the emergence of some variants of concern

(i.e., Omicron) limits their effectiveness. Thus, there is an urgent need to develop antiviral therapeutics that are effective against SARS-CoV-2 and related coronaviruses.

The SARS-CoV-2 genome encodes two cysteine proteases, the 3-chymotrypsin-like protease (3CLPro or Mpro) and the papain-like protease (PLpro), both of which are essential for viral maturation. PLpro is a 35-kDa domain of Nsp3, a 215-kDa multidomain protein that is key to maturation of the viral replicase-transcriptase complex

(RTC)[5]. PLpro cleaves the viral polyproteins pp1a and pp1ab at three sites to produce nonstructural proteins Nsp1, Nsp2, and Nsp3. In addition to RTC maturation, PLpro enables evasion of the host immune response by cleaving ubiquitin and the ubiquitin-like interferon-stimulated gene 15 (ISG15) protein from host protein conjugates[6–8]. Compared to PLpro from SARS-CoV (SARS-CoV PLpro), SARS-CoV-2 PLpro displays decreased deubiquitinase activity and enhanced deISGylation activity[9–11]. In addition, PLpro attenuates type I interferon pathways involved in mediating antiviral immune responses[10]. Inhibition of SARS-CoV-2 PLpro is shown to reduce viral replication in Vero CCL-81 cells[12] and to maintain the host interferon pathway[10].

PLpro consists of thumb, fingers, and palm subdomains common to other ubiquitin-specific proteases, and an N-terminal ubiquitin-like domain involved in substrate recognition (Fig. 1a). The active site, which is located at the interface of the thumb and palm subdomains, consists of a catalytic triad comprising Cys111, His272, and Asp286[12–14]. Besides the catalytic Cys111, four Cys residues coordinate a structural $Zn^{2+}$ cation in the fingers subdomain and six additional Cys residues are present elsewhere in the protein. Of all the cysteines in PLpro, Cys111 is the most prone to oxidation[14], indicating that it is unique in its reactivity toward electrophiles.

Protein substrates of PLpro consist of a Leu-X-Gly-Gly peptide motif (X = Arg, Lys, or Asn), with proteolytic cleavage occurring after the second Gly residue[6]. Leu and X occupy the S4 and S3 sites, respectively, and the two Gly residues occupy the S2 and S1 sites, which are covered by a β-hairpin blocking loop (BL2 loop) that forms a narrow groove leading to the active site (Figs. 1 and 2a)[12]. As a result, only extended peptide substrates with two Gly residues at the P1 and P2 positions can be accommodated in this space[11,12].

Several noncovalent inhibitors of PLpro have been developed that competitively inhibit PLpro[14–17]. The naphthylmethylamine compound GRL0617 inhibits SARS-CoV PLpro with an $IC_{50}$ of ~0.6 μM and inhibits viral replication in Vero E6 cells with $EC_{50} = 14.5$ μM[15]. The desamino analog of GRL0617 exhibits similar inhibitory activity ($IC_{50} = 2.3$ μM; $EC_{50} = 10$ μM), as does the N-acetylated analog ($IC_{50} = 2.6$ μM; $EC_{50} = 13.1$ μM). GRL0617 exhibits similar inhibition activity against SARS-CoV-2 PLpro[10,14,18]. GRL0617 does not inhibit structurally similar human deubiquitinases (DUBs). The $IC_{50}$ values for GRL0617 toward HAUSP, the deISGylase USP18, and the ubiquitin C-terminal hydrolases UCH-L1 and UCH-L3 are all >100 μM[15]. In addition, GRL0617 does not display cytotoxicity at concentrations up to 50 μM in cell viability assays. Crystallographic studies[14,15] have revealed key interactions between PLpro and GRL0617 including (i) a hydrogen bond between the backbone N-H of Gln269 and the amide carbonyl of the inhibitor, (ii) a hydrogen bond between the N-H of the GRL0617 amide and the carboxylate side chain of Asp164, and (iii) an edge-to-face interaction of the naphthyl group of GRL0617 and Tyr268 (Fig. 1b).

Here we design covalent inhibitors of PLpro based on GRL0617. We report in vitro inhibition ($IC_{50}$, $k_{inact}/K_I$), cytopathic protection and virus yield reduction ($EC_{50}$, $EC_{90}$) and cytotoxicity ($CC_{50}$), electrospray ionization mass spectrometry, X-ray crystallography, enzyme selectivity, metabolic stability, and pharmacokinetics data. We show that the most promising candidate is a potent and selective covalent inhibitor of PLpro from SARS-CoV-2.

## Results

We designed a series of covalent PLpro inhibitors based on the noncovalent inhibitor GRL0617 (Fig. 2). Previous crystallographic studies have revealed that the phenylmethyl group of GRL0617 points toward the active site of PLpro but is located >7 Å from Sγ of Cys111 (Fig. 2b)[14]. We reasoned that replacing the methyl substituent of GRL0617 with a hydrolytically stable linker connected to an electrophile capable of reacting with Cys111 would yield a potent covalent inhibitor of PLpro. We chose an N,N'-acetylacetohydrazine linker as a linear Gly-Gly peptidomimetic that could reach through the narrow S2 and S1 groove to the active site while also preserving some of the hydrogen-bonding interactions (e.g., with Gly163 and Gly271) afforded by natural peptide substrates. To the resulting linker we appended a series of electrophiles including a fumarate methyl ester[19], chloroacetamide[20], propiolamide, cyanoacetamide, and α-cyanoacrylamide (Fig. 2c) with the expectation that they would form a covalent adduct with Cys111 (Fig. 2d).

To help prioritize designed molecules for synthesis and testing, we performed covalent docking of each candidate molecule to PLpro (Fig. 3). We also docked each molecule non-covalently to assess the favorability of pre-covalent binding. We used an ensemble of 50 structural models derived from X-ray crystallographic data to account for protein flexibility[14] and included selected crystallographic water molecules during docking, including those that are known to remain bound in the S4 subsite in the presence of noncovalent inhibitors (Supplementary Fig. 1)[14,15]. All candidate inhibitors contain the naphthylmethylamine core of GRL0617 and we aimed for our modified compounds to recapitulate its crystallographic binding mode. To assess pose similarity, we measured the maximum common substructure RMSD (MCS-RMSD) between the docked poses of the candidate inhibitors and the crystallographic pose of GRL0617. In general, the core of the inhibitor designs and their noncovalent precursors reproduced the binding mode of GRL0617 to within 2 Å RMSD, maintaining interactions with Asp164, Tyr268, and Gln269 while the linker occupied the S2 and S1 subsites to place the electrophilic group near the catalytic Cys111 nucleophile (Fig. 3 and Supplementary Fig. 2). Compounds were prioritized for synthesis based on low MCS-RMSD values (≤2 Å), favorable noncovalent and covalent docking scores (Supplementary Fig. 3 and Supplementary Data 1), and synthetic tractability.

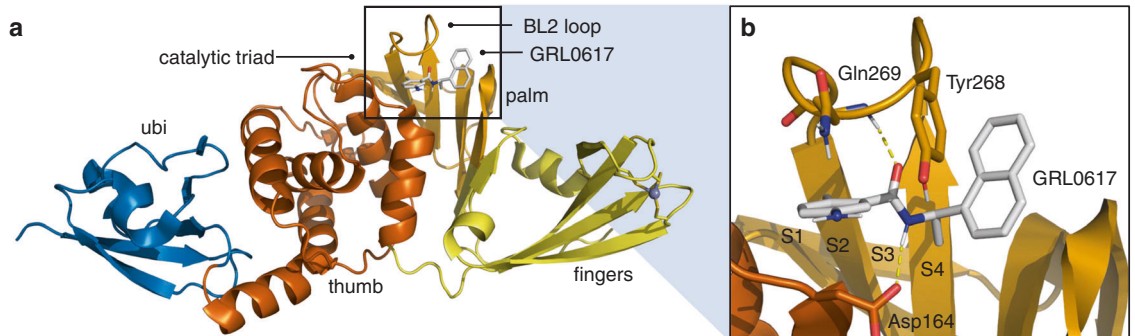

**Fig. 1 | Structure of PLpro from SARS-CoV-2. a** Overall structure (PDB entry 7JIR[14]) colored by domain and selected features labeled. **b** Interactions between PLpro and the noncovalent inhibitor GRL0617. Selected residues and substrate binding subsites are labeled.

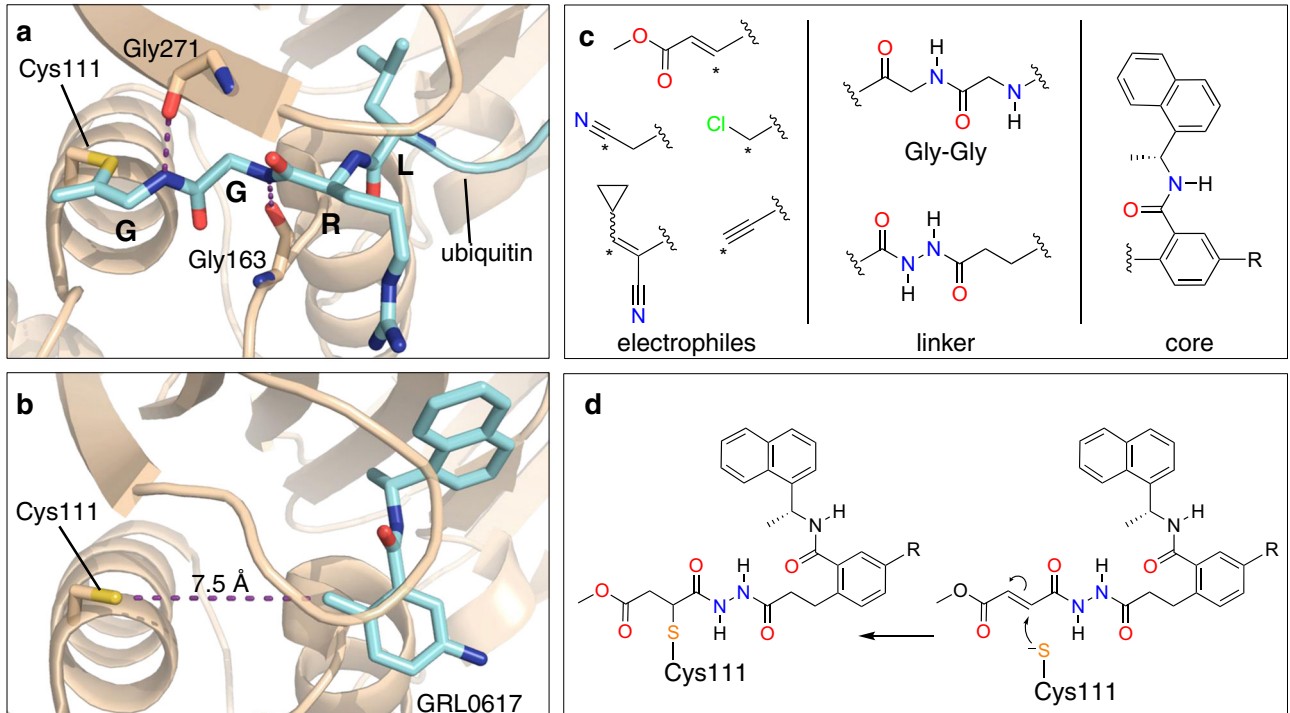

**Fig. 2 | Design strategy for covalent PLpro inhibition. a** X-ray co-crystal structure of ubiquitin-propargylamine (cyan) covalently bound to Cys111 in PLpro (tan) from PDB entry 6XAA[12]. Selected residues from PLpro and the LRGG motif of ubiquitin (cyan) are labeled and shown in stick representation. **b** Crystal structure of GRL0617 (cyan) bound to PLpro (PDB entry 7CMD)[18]. The distance between Sγ of Cys111 and the tolyl methyl of GRL0617 is labeled. **c** Components of covalent PLpro inhibitor candidates consisting of various electrophiles, a Gly-Gly mimetic linker, and the GRL0617 core. Reactive carbons on electrophiles are labeled with asterisks. **d** Mechanism of covalent bond formation between Cys111 and an inhibitor candidate with a fumarate ester electrophile.

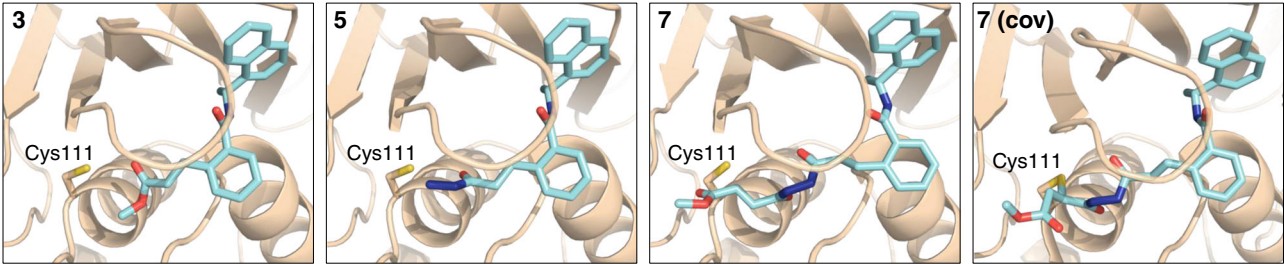

**Fig. 3 | Docked poses of compounds 3, 5, and 7.** Compound **7** was docked both noncovalently and covalently. Structures of compounds are shown in Fig. 4. Docked poses for additional inhibitor candidates are shown in Supplementary Fig. 2. Ligand carbons are shown in cyan. Hydrogens were omitted for clarity.

We synthesized compounds **2**–**15** beginning from an amide coupling of (R)-( + )-1-1-(1-napthyl)ethylamine and 2-(3-methoxy-3-oxopropyl) benzoic acid derivatives, where $R_1$ = H or NHAc (Fig. 4). Following this coupling, we reacted the ester in **3** and **4** with $N_2H_4 \cdot H_2O$ in refluxing EtOH to afford the hydrazide group in **5** and **6** in near quantitative yield. With the respective hydrazides in hand, we installed a variety of electrophiles using acid chlorides. The solubility of **5** and **6** were quite different from each other and required separate conditions for installation of the electrophilic groups. DIPEA/DCM was used for **5** ($R_1$ = H) and $K_2CO_3$/DMF was used for **6** ($R_1$ = NHAc). Overall, we synthesized seven covalent inhibitor candidates (**7**-**13**) and two additional noncovalent GRL0617 derivatives, namely compounds **14** ($R_1$ = H) and **15** ($R_1$ = NHAc).

The synthesized compounds were then assessed for potential anti-SARS-CoV-2 activity in a biochemical assay using purified PLpro and the ubiquitin C-terminus-derived fluorogenic substrate Z-RLRGG-AMC[15,21,22] (Table 1 and Supplementary Fig. 4). IC_{50} values were determined following a 30-minute incubation of PLpro with inhibitor (Supplementary Fig. 5). Of the noncovalent analogs of GRL0617, we found that both **14** and **15** had increased IC_{50} values, with the N-acetylated compound **15** having an IC_{50} more like that of GRL0617 (Table 1). Extending the tolyl methyl to include a larger peptidomimetic group did not adversely affect potency. For example, addition of the linker alone without an electrophile to form **5** led to an IC_{50} of 24 µM (Supplementary Fig. 5). The introduction of five different electrophilic groups to produce compounds **7**, **9**, and **11**–**13** resulted in improved IC_{50} values for all except α-cyanoacrylamide **13**. Time-dependent inhibition assays were performed because time-dependence is consistent with multiple mechanisms of slow-binding inhibition, including covalent inhibition via bond formation between Cys111 and the electrophile. Installation of a chloroacetamide electrophile to form **9** improved the IC_{50} compared to **5** to 5.4 µM after 30-min incubation and resulted in a $k_{inact}/K_I$ of 110 M$^{-1}$ s$^{-1}$ (Supplementary Fig. 6), where $k_{inact}/K_I$ is a second-order rate constant describing the efficiency of the overall conversion of free enzyme to the covalent enzyme-inhibitor complex[23]. Similarly, the IC_{50} and $k_{inact}/K_I$ for N-acetylated analog **10** are 4.4 µM and 140 M$^{-1}$ s$^{-1}$, respectively.

**Fig. 4 | Synthesis of compounds 2-15.** Reaction conditions with yields in parentheses: I. Ac$_2$O, AcOH, DCM, 55%; II. HATU, DIPEA, DCM (**3**, 83%; **4**, 91%); III. N$_2$H$_4$•H$_2$O, EtOH (**5** and **6**), 97%); IV. methyl (E)-4-chloro-4-oxobut-2-enoate, DIPEA, DCM for **7** (56%), and K$_2$CO$_3$, DMF for **8** (34%). Compounds **9** (50%), **10** (37%), **11** (56%), **12** (23%), and **13** (60%) were prepared with the corresponding acid chlorides under conditions described for step IV. Compounds **14** (89%) and **15** (83%) were prepared analogously to step II with 2-methylbenzoic acid and 5-acetamido-2-methylbenzoic acid, respectively.

### Table 1 | PLpro inhibition and SARS-CoV-2 antiviral activity

| Compound | $R_1$[a] | Electrophile | IC$_{50}$ (µM)[b] | Time dep. | $k_{inact}/K_I$ (M$^{-1}$ s$^{-1}$) | EC$_{50}$ (µM)[c] | CC$_{50}$ < 30 µM |
|---|---|---|---|---|---|---|---|
| GRL0617 | NH$_2$ | NA | 1.2 | No | NA | ND | ND |
| **3** | H | NA | >100 | No | NA | ND | ND |
| **5** | H | NA | 24 | No | NA | ND | ND |
| **7** | H | Fumarate ester | 0.094 | Yes | 9,600 | 1.1 | No |
| **8** | NHAc | Fumarate ester | 0.230 | Yes | 9,000 | No CPE | No |
| **9** | H | Chloroacetamide | 5.4 | Yes | 110 | 34 | No |
| **10** | NHAc | Chloroacetamide | 4.4 | Yes | 140 | No CPE | No |
| **11** | H | Cyanoacetamide | 8.0 | No | ND | No CPE | No |
| **12** | H | Propiolamide | 0.098 | Yes | 4,100 | No CPE | Yes |
| **13** | H | α-cyanoacrylamide | >200 | No | ND | No CPE | Yes |
| **14** | H | NA | 100 | ND | NA | No CPE | No |
| **15** | NHAc | NA | 6.2 | ND | NA | No CPE | No |

NA not applicable, ND not determined.
[a]Structures are shown in Fig. 4.
[b]Measurement after 30-min incubation. Purified PLpro with Z-RLRGG-AMC substrate.
[c]Cytopathic effect in SARS-CoV-2-infected Vero E6 cells. EC$_{50}$ for remdesivir = 0.74 µM.

Based on previous success in incorporating a vinyl methyl ester electrophile into tetrapeptide-based, irreversible covalent inhibitors of PLpro[11], we reasoned that incorporating a similar ester into our covalent inhibitor candidates would occupy the oxyanion hole in the active site and that the ester carbonyl oxygen would engage in a hydrogen bond with the indole N-H of Trp106. Fumarate methyl ester **7** had an IC$_{50}$ of 0.094 µM after 30-min incubation and $k_{inact}/K_I$ = 9600 M$^{-1}$ s$^{-1}$, indicating potent inhibition (Table 1 and Supplementary Fig. 6). N-acetylated analog **8** showed similar potency, with IC$_{50}$ and $k_{inact}/K_I$ = 0.23 µM and 9000 M$^{-1}$ s$^{-1}$, respectively. To examine the inhibitory activity of other electrophiles, we synthesized and performed time-independent inhibition assays with cyanoacetamide **11** (IC$_{50}$, 8 µM), propiolamide **12** (0.098 µM), and α-cyanoacrylamide **13** (>200 µM). Time-dependent inhibition was observed for **12** (Supplementary Fig. 6), but not for **11** or **13** (Supplementary Fig. 7). To provide additional evidence for a covalent mechanism of action, compounds **7-10** and **12** were incubated with PLpro, and the protein intact masses were

determined by electrospray ionization mass spectrometry (ESI-MS). Covalent adduct formation with PLpro was confirmed for these five compounds (Fig. 5c, Supplementary Fig. 8, and Supplementary Table 1).

We next assessed the ability of selected inhibitors to protect Vero E6 cells from virus-induced cell death, represented by EC$_{50}$ (Table 1, Fig. 5d, and Supplementary Fig. 9), by incubating cells with and without compound and then infecting them with SARS-CoV-2[24]. Uninfected cells were used to assess the cytotoxicity of the compounds, represented by CC$_{50}$. Compound **7** displayed notable antiviral activity with an EC$_{50}$ of 1.1 µM, comparable to that of the remdesivir positive control (0.74 µM). Chloroacetamide **9** also displayed antiviral activity, although with less potency (34 µM). Neither **7** nor **9** displayed evidence of cytotoxicity (CC$_{50}$ > 30 µM). Compounds **8** and **10**, which have N-acetylated phenyl substituents, showed insignificant cytoprotective effects. Both **12** and **13** were cytotoxic with CC$_{50}$ values of 1–5 µM, suggesting that propiolamide

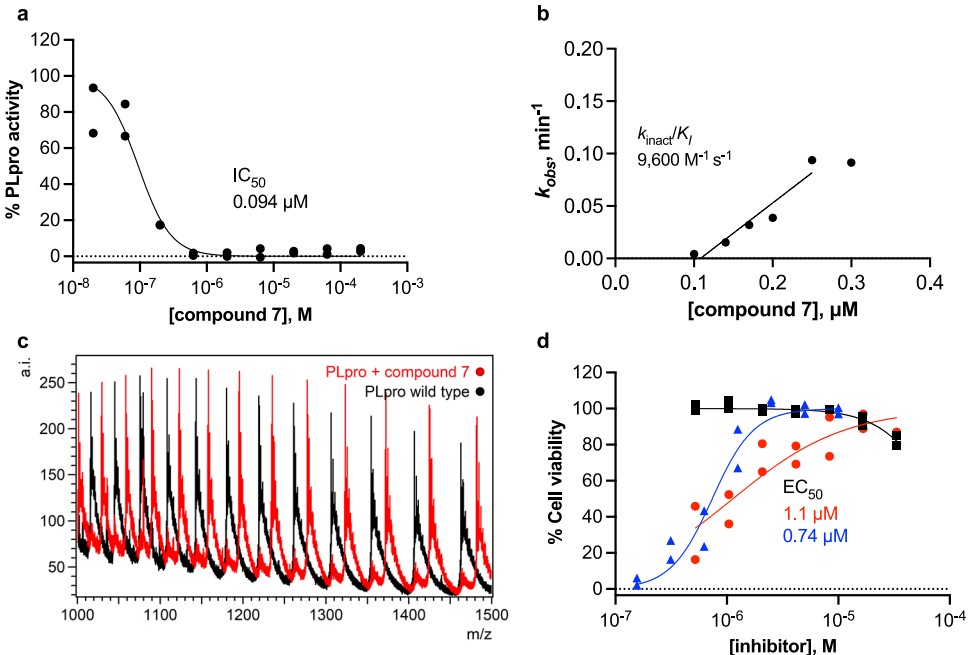

**Fig. 5 | Characterization of a designed covalent PLpro inhibitor, compound 7.**
**a** Fluorogenic peptide activity assay after 30-min preincubation with compound **7**. Data are plotted for each of $n = 2$ independent samples. IC$_{50}$ is the concentration at which 50% inhibition was observed. Curve is the nonlinear regression to the normalized inhibitor dose response equation. **b** Time-dependent characterization with a fluorogenic peptide assay. Data points are $k_{obs}$ values determined by fitting the exponential decay equation to initial rates determined at various inhibitor concentrations and preincubation times, normalized to no preincubation. $k_{obs}$ data are presented as mean values determined from $n = 2$ independent samples. Line represents the linear regression yielding as its slope the second-order rate constant ($k_{inact}/K_I$). **c** Intact protein ESI-MS spectra of PLpro (black) and PLpro incubated with **7** (red); a.i., arbitrary intensity; m/z, mass-to-charge ratio. **d** Percent viability of Vero E6 cells after 48 h following pretreatment with **7** (black squares), pretreatment with **7** and infection with SARS-CoV-2 (red circles), or pretreatment with remdesivir and infected with SARS-CoV-2 (blue triangles). Data are plotted as the mean of $n = 2$ independent samples. EC$_{50}$ is the concentration at which 50% effect was observed. Curves are nonlinear regressions to the normalized dose response equation. Source data are provided as a Source Data file.

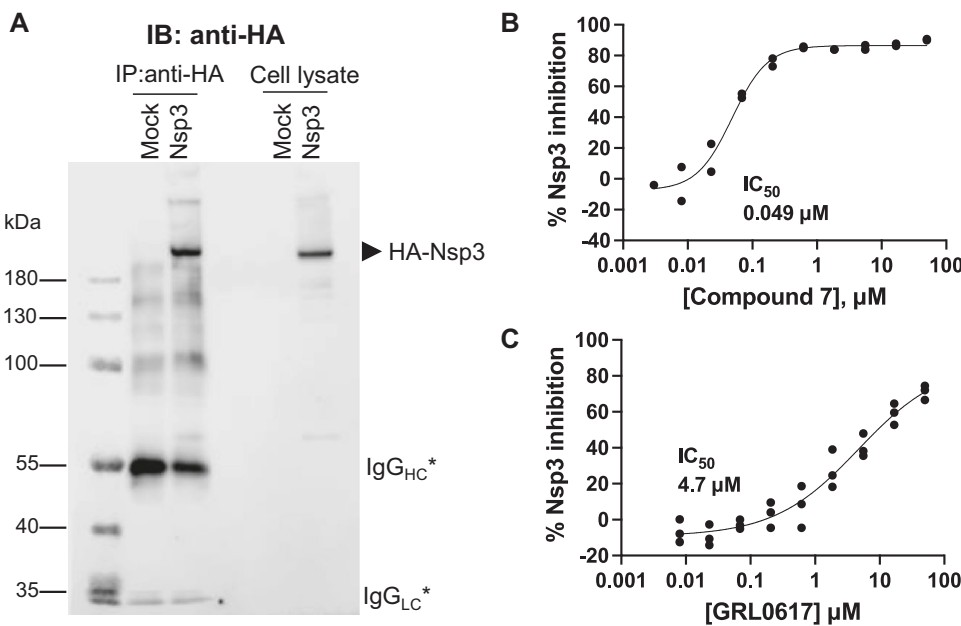

**Fig. 6 | Inhibition of the deISGylase activity of full-length SARS-CoV-2 hemagglutinin (HA)-Nsp3 transiently expressed in HEK293T cells. A** Anti-HA beads after immunoprecipitation (IP) and whole cell lysates probed with anti-HA antibody. The asterisks indicate immunoglobulin G (IgG) heavy chain (HC) and light chain (LC). Anti-HA beads were assayed for Nsp3 deISGylase activity using an ISG15-CHOP2 assay in the presence of the dose range of **B** compound **7** or **C** GRL0617. Data are presented as mean values for $n = 2$ independent experiments for compound **7** and $n = 3$ independent samples for GRL0617. Curves are nonlinear regressions to the normalized dose response equation. Source data are provided as a Source Data file.

**Table 2 | Cytopathic effect of compound 7 against three variants of SARS-CoV-2 in Vero E6 cells in the presence of 2 µM CP-100356**

| Strain | Compound | $EC_{50}^{a}$ (µM) | $CC_{50}^{b}$ (µM) | $SI_{50}^{c}$ |
|---|---|---|---|---|
| USA-WA1/2020 | **7** | 0.068 | >10 | >150 |
| | EIDD-1931 | 0.3 | >100 | >330 |
| Delta (B.1.617.2) | **7** | 0.29 | >10 | >34 |
| | EIDD-1931 | 0.31 | >100 | >320 |
| Omicron (B.1.1.529) | **7** | 0.68 | >10 | >15 |
| | EIDD-1931 | 0.3 | >100 | >330 |

The RNA-dependent RNA polymerase inhibitor EIDD-1931 was used as a positive control.
$^{a}EC_{50}$ = 50% effective concentration.
$^{b}CC_{50}$ = 50% cytotoxic concentration.
$^{c}SI_{50}$ = $CC_{50}/EC_{50}$.

**Table 3 | Virus Yield Reduction Data for Compound 7 Against Three Variants of SARS-CoV-2 in Caco-2 Cells**

| Strain | Compound | $EC_{90}^{a}$ (µM) | $CC_{50}^{b}$ (µM) | $SI_{90}^{c}$ |
|---|---|---|---|---|
| USA-WA1/2020 | **7** | 0.26 | >10 | >38 |
| | EIDD-1931 | 0.12 | 94 | 780 |
| Delta (B.1.617.2) | **7** | >10 | >10 | 0 |
| | EIDD-1931 | 4.9 | >100 | >20 |
| Omicron (B.1.1.529) | **7** | 2.4 | >10 | >4.2 |
| | EIDD-1931 | 2.9 | >100 | >34 |

The RNA-dependent RNA polymerase inhibitor EIDD-1931 was used as a positive control.
$^{a}EC_{90}$ = 90% effective concentration.
$^{b}CC_{50}$ = 50% cytotoxic concentration.
$^{c}SI_{90}$ = $CC_{50}/EC_{90}$.

and α-cyanoacrylamide electrophiles were too reactive, lack specificity, or both.

In addition to its role in processing the replicase polyprotein, SARS-CoV-2 PLpro displays deubiquitinase and de-ISG15ylase activity[12,25]. To ensure that the most promising covalent inhibitors, **7** and **9**, can inhibit these physiologically relevant activities, $IC_{50}$ values were obtained with Ub-rhodamine and ISG15-CHOP2 substrates (Supplementary Table 2). Compound **7** inhibited PLpro with Ub-rhodamine and ISG15 substrates with $IC_{50}$ values of 0.076 and 0.039 µM, respectively. The corresponding $IC_{50}$ values for **9** with these two substrates were 1.96 µM and 20.2 µM, respectively. We then performed selectivity assays with **7** and **9** against a panel of seven human DUBs: USP2c, USP4, USP7, USP8c, USP15, USP30WT, and UCH-L1. Neither compound inhibited any of the seven human DUBs tested ($IC_{50}$ > 30 µM in all cases), indicating selectivity toward PL$^{pro}$.

Although small molecule-mediated inhibition has been reported for recombinant PLpro domain and for truncated Nsp3[26], direct inhibition of full-length Nsp3 has not yet been demonstrated. Thus, we expressed full-length hemagglutinin (HA)-Nsp3 in HEK293T cells and purified the enzyme using anti-HA immunoprecipitation (Fig. 6). We found that compound **7** potently inhibited the deISGylase activity of full-length Nsp3 ($IC_{50}$ = 0.049 µM). In contrast, GRL0617 showed much weaker inhibition ($IC_{50}$ = 4.7 µM) under the same assay conditions.

To assess the efficacy of **7** against various SARS-CoV-2 strains, we performed CPE assays with Vero E6 cells infected with the USA-WA1/2020, Delta (B.1.617.2), or Omicron (B.1.1.529) variant (Table 2). Vero cells overexpress the efflux transporter P-glycoprotein (P-gp), so we performed these assays of **7** in the presence of the P-gp inhibitor CP−100356[27]. Using a neutral red staining assay, we observed variant-dependent $EC_{50}$ values of 0.068 µM for USA-WA1/2020, 0.29 µM for Delta, and 0.68 µM for Omicron.

To assess the antiviral activity of **7** in human cells, we evaluated the compounds in virus yield reduction assays using Caco-2 cells. We measured $EC_{90}$ values for **7** in Caco-2 cells infected with the USA-WA1/2020, Delta (B.1.617.2), or Omicron (B.1.1.529) variant (Table 3). In contrast to the cytopathic protection assays performed with Vero E6 cells, the results varied more among strains in this case. The $EC_{90}$ was 0.26 µM for USA-WA1/2020, >10 µM for Delta, and 2.4 µM for Omicron.

Following the promising results from in vitro assays and mass spectrometry experiments, we determined a crystal structure of wild-type PLpro in complex with **7** at 3.10 Å resolution (Supplementary Table 3). The electron density maps show clear densities for PLpro, Zn cations, and **7**, confirming the design concept of this compound and revealing key interactions with PLpro (Fig. 7). A covalent bond is present between Sγ of Cys111 and the β carbon of the ester of **7** (Fig. 7a, c). The carbonyl oxygen of the ester accepts hydrogen bonds from the indole side chain of Trp106, like that of the tetrapeptide-based covalent inhibitor VIR251[11], and also from the side chain of Asn109. The *N,N*'-acetylacetohydrazine moiety was designed to link the electrophile

and the naphthylmethylamine core while also hydrogen bonding with residues in the S1-S2 groove. Indeed, the crystal structure revealed that the proximal and distal carbonyl oxygens of the linker interact with the backbone N-H groups of Gly163 and Gly271, and the proximal and distal N-H groups of this moiety participate in hydrogen bonds with the carbonyl backbones of Gly271 and Gly163. As intended, the carbonyl oxygen and N-H group of the amide adjacent to the naphthyl group of **7** are hydrogen bonded with the N-H backbone of Gln269 and the carboxylate side chain of Asp164. Compound **7** makes five main-chain and three side-chain hydrogen bonding interactions in the binding site. In addition, the side chains of Tyr268 and Gln269 interact with **7** similarly to GRL0617. Electron density for the methyl group of the ester of **7** was not visible. It is possible that the ester linkage is flexible and adopts multiple conformations or that it could have been hydrolyzed after covalent bond formation. Encouragingly, the covalently docked pose for **7** agrees closely with the co-crystal structure (Supplementary Fig. 10).

To characterize the binding mode of **7** and the arrangement of residues in the inhibitor binding site, we overlayed crystal structures of PLpro with and without bound ligands. In general, the conformational changes in the residues around the binding site are relatively small among the different PLpro structures. However, Leu162 and the BL2 loop in particular display substantial movement between the unbound and bound structures (Fig. 7d). In the unbound structure, Leu162 adopts a closed conformation in which the side chain of Leu162 folds inward toward the catalytic groove and blocks access to the catalytic Cys111[14]. This closed conformation of Leu162 is also present in co-crystal structures of PLpro with GRL0617 (Fig. 7e) and other inhibitors that do not extend into the S2 and S1 pockets (Supplementary Fig. 11)[18,28]. In these instances, the side chain of Leu162 may be stabilized by hydrophobic interactions with the inhibitor and other residues around the pocket. In contrast, in the co-crystal structure of PLpro with **7** the side chain of Leu162 is rotated outward away from the BL2 loop to allow the electrophile to access Cys111. This outward rotation of the Leu162 side chain in the presence of the inhibitor is similar to other structures with peptides or small-molecule inhibitors that extend more deeply into the catalytic groove (Supplementary Fig. 11)[11,29]. The BL2 loop, which partially covers the substrate binding groove, undergoes a large conformational change upon binding of **7**. In the unbound structure[14], the side chains of residues forming this loop face outward and expose the groove. However, when **7** is bound the BL2 loop shifts inward and covers the inhibitor to stabilize the bound form (Fig. 7d). Similar conformational changes of the BL2 loop have also been observed in co-crystal structures of other inhibitors including GRL0617 (Supplementary Fig. 11)[18,28].

To help rationalize our findings from the DUB selectivity assays, we superimposed the co-crystal structure of **7** bound to PLpro onto the structure of human deubiquitinase UCH-L1[29]. The crossover loop of UCH-L1 (residues 153–157) overlaps with the narrow substrate-binding

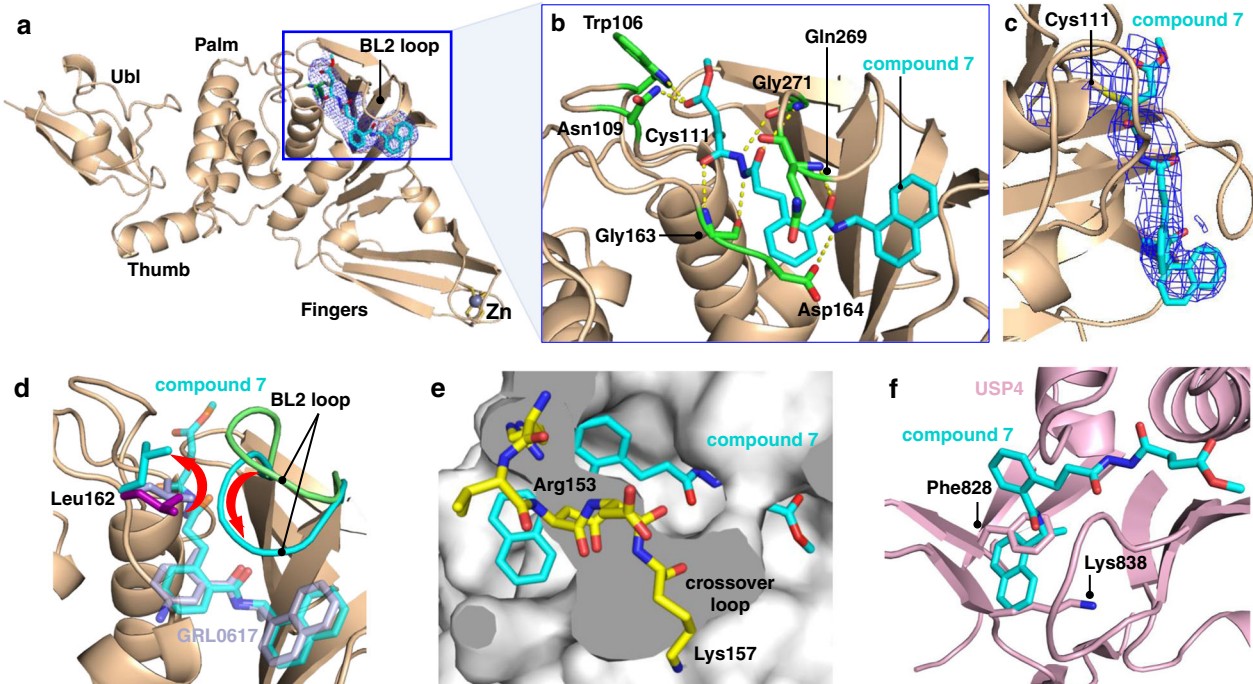

**Fig. 7 | Crystal structure of SARS-CoV-2 PLpro in complex with covalent inhibitor 7. a** Overall structure. The electron density for **7** is shown in blue mesh (Fo - Fc omit map contoured at 1.5 σ). **b** Interactions between binding site residues (green sticks) and **7** (cyan sticks). **c** Composite omit map (σ = 1.0) showing the electron density for the covalent bond between Cys111 and **7**. **d** Superposition of selected structures highlighting the positions of the side chain of Leu162 (sticks) and the BL2 loop (cartoon) in the absence and presence of selected inhibitors: Ligand-free (PDB entry 6W9C, light green), glycerol-bound (PDB entry 6WZU, purple), GRL0617-bound (PDB entry 7CMD, light purple), and compound **7**-bound (this work; cyan).

Additional structures are shown in Supplementary Fig. 11. **e** Structural basis for selectivity toward PLpro. Superposition of **7** bound to PLpro onto human carboxy terminal hydrolase UCH-L1[29] (PDB entry 3KW5). The crossover loop of UCH-L1 (residues 153–157) covers the narrow groove and likely blocks the naphthyl-methylamine core of **7** from binding. **f** Superposition of **7** bound to PLpro onto human USP4[31] (PDB entry 2Y6E). Severe steric clashes are present between the naphthyl ring of **7** and Phe828 and Lys838 of USP4 (light pink sticks), both of which are conserved in 80% of human USPs.

groove in PLpro, which likely prevents **7** from binding (Fig. 7e). In UCH-L3 and UCH-L5, however, the crossover loop is longer and, in some cases, more disordered[30]. Thus, there is sufficient space for **7** to bind to these proteins, although it has been shown previously that GRL0617 does not inhibit UCH-L3[15]. We next superimposed the co-crystal structure of **7** bound to PLpro onto human USP4[31]. Severe steric clashes are present between the naphthyl ring of **7** and Phe828 and Lys838 of USP4 (Fig. 7f), both of which are conserved in 80% of human USPs. These findings suggest that structural analyses and virtual counter screens of candidate inhibitors against cysteine proteases from the human proteome may be useful in identifying compounds that are less susceptible to off-target binding.

To assess the in vitro ADME properties of **7**, **9**, and **14**, we determined their metabolic stabilities in human, rat, and mouse liver microsomes and the corresponding S9 fractions (Supplementary Tables 4 and 5). Compound **14** was selected to enable a direct comparison of the two most promising covalent inhibitor candidates with a reference noncovalent inhibitor that lacks the linker and electrophile. In mouse liver microsomes and S9 fraction, **14** had half-lives of 16 and 18 min, respectively. Fumarate methyl ester **7** exhibited somewhat shorter half-lives of 13 min in mouse liver microsomes and 9 min in the S9 fraction. Chloroacetamide **9** demonstrated very short half-lives of 5 and 4 min in microsomes and S9 fractions, respectively. In human liver microsomes and S9 fraction, the half-lives of **14** were 41 min and >60 min. Similarly, **7** exhibited a half-life of 50 min in microsomes and 60 min in the S9 fraction, indicating that the additional liabilities introduced by the linker and fumarate ester electrophile are relatively minor. In contrast, chloroacetamide **9** demonstrated very short half-lives of 7 and 3 min in human liver microsomes and S9 fractions, respectively, apparently due to the chloroacetamide electrophile.

Analysis of **7** and **14** with MetaSite 6.0.1[31] was performed to predict metabolic transformations from CYP450s and flavin-containing monooxygenase in phase 1 metabolism (Supplementary Fig. 12). The results for **14** suggested that the chiral carbon and tolyl methyl are the predominant metabolic liabilities. For **7** the linker and electrophile replaced the tolyl methyl, so it is unsurprising that the resulting benzylic methylene is predicted to be a primary site of metabolism. An additional liability for **7** is predicted to be the methyl ester. The naphthyl also has several predicted metabolic hotspots in both **7** and **14**. We advanced **7** into a pharmacokinetic study to assess its in vivo exposure. Male ICR mice were dosed with 10 mg/kg (p.o.) or 3 mg/kg (i.v.) to obtain a more complete picture of the PK/PD profile. Unfortunately, **7** was not orally bioavailable and there was no exposure recorded following PO dosing. The half-life ($t_{1/2}$) following IV dosing was 0.06 h and the clearance was 11,047 mL/min/kg. Additional PK parameters are summarized in Supplementary Table 6. Little exposure was observed and the levels of **7** did not meet the threshold for progression into an in vivo efficacy study. Based on the in vitro ADME results, the PK findings were unsurprising. Taken together, they highlight key areas for the development of next-generation covalent PLpro inhibitors.

## Discussion

Numerous research efforts have focused on developing inhibitors of 3CLpro, but very few successful efforts have been reported on PLpro inhibition as it is difficult to drug because of its featureless and flexible binding pockets. Another reason for the emphasis on 3CLpro as an antiviral target is that there are no structural homologs in the human proteome, whereas PLpro bears structural similarity to human DUBs and deISGylases. Nevertheless, PLpro is a promising target for

developing therapeutics to combat SARS-CoV-2. Recent reviews have highlighted the increased interest in and progress toward more effective PLpro inhibitors[29,32].

In the present work we have pursued a computational, biochemical, and structural approach to design covalent inhibitors based on the most well-studied noncovalent inhibitor of PLpro, GRL0617. We have developed a covalent PLpro inhibitor that improves upon the potency of its parent noncovalent inhibitor. Acknowledging the limitations of $IC_{50}$ measurements for covalent inhibitors[23], the addition of a linker and electrophile to the GRL0617 core improved $IC_{50}$ by ~100-fold for **7** with Nsp3 and ISG15 substrate. Whereas some of our candidate inhibitors are attacked at the α carbon (**7–10**), others are attacked at the β carbon (**11–13**). The general trend from in vitro inhibition assays suggests that the α carbon is at the right distance and geometry to react with Cys111. Furthermore, a crystal structure of **7** covalently bound to PLpro provides structural insights that will facilitate the development of next-generation covalent PLpro inhibitors. Encouragingly, **7** exhibited $k_{inact}/K_I$ = 9600 $M^{-1}$ $s^{-1}$ with PLpro and peptide substrate, and low-μM $EC_{50}$ values in mammalian cells infected with SARS-CoV-2 virus. Cytopathic effect assays of **7** performed with Vero E6 cells and virus yield reduction assays with Caco-2 cells revealed somewhat weaker activity against the B.1.617.2 (Delta) or B.1.1.529 (Omicron) variants compared to the USA-WA1/2020 strain but the reason for this observation is unclear. However, we note that there are no characteristic mutations in the PLpro domain of Nsp3 from the B.1.617.2 or B.1.1.529 variants relative to USA-WA1/2020.

Future goals for designing improved covalent inhibitors of PLpro should emphasize improving in vitro ADME and in vivo PK/PD properties, while simultaneously optimizing potency, selectivity, solubility, and permeability. Encouragingly, the difference in metabolic stability for covalent inhibitor **7** compared to noncovalent parent compound **14** were small, particularly in human liver microsomes. However, the parent noncovalent inhibitor already exhibits major liabilities and compounds based on it inherit these liabilities. To address these liabilities several modifications could be pursued. Although naphthyl groups are present in some pharmaceutical compounds, they are highly susceptible to metabolic degradation and are also considered to be toxicophores[33]. A recent report of non-covalent PLpro inhibitors based on GRL0617 showed that replacing the naphthyl group with substituted 2-phenylthiophenes yielded inhibitors that mimic the binding interaction of ubiquitin with Glu167 of PLpro, simultaneously improving $IC_{50}$ values and metabolic stability[28]. Although thiophenes are known to be sensitive to CYP450-mediated metabolism, this vulnerability can be attenuated through substitution with a methyl or halogen substituent at the C-4 or C-5 positions[34]. Modifications at this site would also enable modulation of solubility while maintaining or enhancing local interactions with the BL2 loop. In addition, male C57BL/6 mice dosed with 50 mg/kg (i.p. injection) of XR8-23 and XR8-24, 2-phenylthiophene-containing derivatives of GRL0617, were estimated to reach plasma concentrations of ~12 μM, indicating promising bioavailability. Thus, replacement of the naphthyl substituent in analogs of **7** should provide clear benefits.

Substitution of benzylic and other nonpolar hydrogens in **7** with fluorine[35,36] or deuterium[37] is a common strategy for reducing oxidative metabolism. The introduction of fluorine into compounds has been associated with decreased clearance and increased permeability, both of which could lead to increased exposure in vivo. To increase steric hindrance[38] and block the site of metabolism, the benzylic methylene in **7** could be replaced with cyclopropyl[39].

The choice of the electrophile is clearly crucial for efficacy, selectivity, and stability. Chloroacetamides are the most used haloacetamide and have demonstrated comparable stability to glutathione at pH 7.4 as α,β-unsaturated amides[40]. In addition, the reactivity can be modulated by introducing steric bulk in proximity to the electrophile[41]. The methyl ester of the fumarate electrophile is labile and its loss

through hydrolysis will lead to inactivation of the electrophile[19,41]. Other ester substituents such as *t*-butyl could be used to tune the kinetics of ester hydrolysis and potentially enhance selectivity by limiting off-target binding[42]. Additional electrophiles for candidate inhibitors include, for example, substituted acrylamides, substituted propiolamides, alkenyl- or alkynyl-substituted heteroarenes, and substituted α-cyanoacrylamides[41,43]. Motivations for these choices are that they are among the most common electrophiles in approved covalent drugs and they have variable substituents that allow for tunable electrophilicity[44] and protein complementarity. Cyanoacrylamides provide the additional benefit that their thiol adducts are reversible, which can reduce the effects of off-target binding[45].

Covalent inhibition is a viable strategy for targeting cysteine proteases that offers advantages over noncovalent inhibition including increased target affinity, lower dose requirements due to longer residence time on target[46,47], lower sensitivity to pharmacokinetic parameters, and lower susceptibility to drug resistance[48–50]. We envision that this mode of action could potentially be targeted for use in combination therapies with drugs targeting 3CLpro or RNA-dependent RNA polymerase. Exploration of inhibitor bioconjugates such as Fc-fusions or E3 ligase fusions is also warranted.

## Methods

### Ethical statement
This research complies with all relevant ethical regulations. All aspects of this work, including housing, experimentation, and disposal of animals were performed in general accordance with the Guide for the Care and Use of Laboratory Animals: Eighth Edition (National Academy Press, Washington, D. C., 2011) in an AAALAC-accredited laboratory animal facility. The animal care and use protocol was reviewed and approved by the IACUC at Pharmacology Discovery Services Taiwan, Ltd. PK profiling assays were performed by Eurofins Panlabs (St. Charles, MO, USA).

### Docking preparation
The 2.09 Å X-ray co-crystal structure of the C111S mutant of PLpro with GRL0617 (PDB entry 7JIR)[14] was used for the docking calculations. Rather than docking to a single structure, we used Phenix[51] to generate an ensemble[52] of 50 conformations from the corresponding crystallographic data in which conformations were sampled to generate an ensemble that collectively fit the data better than any single model. This approach provides valuable information about regions of high and low conformational variability in the protein, such as the BL2 loop, which is known to undergo large conformational changes upon substrate or inhibitor binding. Ser111 was converted back to Cys in all models.

Selected water molecules present in the models were retained during docking. Cys111 was modeled as a neutral thiol and His272 was protonated on Nε in accordance with its local hydrogen bonding environment and the proton transfer chemistry that is expected to occur during catalysis. Other histidines were protonated based on their inferred hydrogen bonding patterns. All other residues were protonated according to their canonical pH 7.0 protonation states. The program *tleap* from AmberTools20[53] was used to prepare the parameter and coordinate files for each structure. The ff14SB force field[54] and TIP3P water model[55] were used to describe the protein and solvent, respectively. Energy minimization was performed using *sander* from AmberTools20 with 500 steps of steepest descent, followed by 2000 steps of conjugate gradient minimization. Harmonic restraints with force constants of 200 kcal $mol^{-1}$ $Å^{-1}$ were applied to all heavy atoms during energy minimization.

The peptide substrate binding cleft of PLpro spans ~30 Å along the interface of the palm and thumb domains (Supplementary Fig. 1). Thus, we defined a rectangular docking box spanning the entire binding cleft (S1–S4 subsites) and the active site (catalytic triad).

AutoGrid Flexible Receptor (AGFR)[56] was used to generate the receptor files for both noncovalent and covalent docking using a grid spacing of 0.25 Å. All docking calculations were performed with AutoDock Flexible Receptor (ADFR)[56]. Compounds with electrophilic groups were docked both noncovalently (i.e., in the reactive form with an explicit electrophile present) and covalently (i.e., in the post-reactive Cys111 adduct form).

## Ligand preparation

SMILES strings for candidate inhibitor designs were converted to PDB format using Open Babel 2.4.1[57] and Python/RDKit[58] scripts. Covalent docking with AutoDockFR requires that ligands be modified such that they include the covalent linkage to the side chain of the reactive residue, in this case Cys111, which then serves as an anchor to place the ligand approximately in the binding site[56]. Thus, the Cα and Cβ atoms of Cys111 were used as anchors and the backbone N atom of Cys111 was used to define a torsional angle connecting the covalently bound ligand and the protein. MGLTools 1.5.6[59] was used to generate PDBQT files for ligands and receptors. Only polar hydrogens were retained during docking.

All candidate inhibitors considered in this work include the naphthylmethylamine core of GRL0617, for which co-crystal structures are available[14]. We expected that our covalent compounds would adopt a pose like GRL0617. Thus, to assess the similarity between the poses of docked candidate ligands and GRL0617 in the X-ray structure, we calculated the maximum common substructure (MCS) RMSD between them. MCS-RMSDs were calculated for all poses with docking energies within 3 kcal/mol of the overall most favorable pose for each candidate inhibitor. Compounds were prioritized for synthesis that had docked poses with MCS-RMSD values ≤2 Å and favorable noncovalent and covalent docking scores (Supplementary Fig. 3 and Supplementary Data 1). Figures were generated with PyMOL[60].

## Synthesis and characterization of compounds

All reagents were purchased from commercial suppliers and used as received. Anhydrous acetonitrile (MeCN), dichloromethane (CH₂Cl₂), ethanol (EtOH), dimethylformamide (DMF), tetrahydrofuran (THF), methanol (MeOH), and diethyl ether (Et₂O) were purchased from commercial sources and maintained under dry N₂ conditions. Amide couplings and reactions with acid chlorides were performed under N₂ using standard Schlenk-line techniques. Compound **1** was purchased from commercial sources and used as received. ¹H and ¹³C NMR spectra were recorded in the listed deuterated solvent with a Bruker Avance III HD 500 MHz NMR spectrometer at 298 K with chemical shifts referenced to the residual protio signal of the deuterated solvent as previously reported[61]. Low-resolution mass data were collected on an Agilent 6470AA Triple Quadrupole LC/MS system. High-resolution mass data were collected on a Waters Synapt HDMS QTOF mass spectrometer. Following the initial synthesis and screening of compounds **2-15**, compound **7** was synthesized at gram scale following the same procedures described below. Purity was analyzed by analytical HPLC and Thermo LTQ MS with electrospray ionization in the positive mode with a Waters BEH 130, 5 μm, 4.6 × 150 mm C18 column, linear gradient from 90:10 to 0:100 water/acetonitrile in 10 min at a flow rate of 1 mL/min. LC/MS chromatograms, ¹H NMR spectra, and ¹³C NMR spectra for all synthesized compounds are provided in Supplementary Figs. 13−52.

**5-acetamido-2-(3-methoxy-3-oxopropyl)benzoic acid (2).** To a 15 mL solution of DCM was added 0.300 g (1.344 mmol) of 5-amino-2-(3-methoxy-3-oxopropyl)benzoic acid and cooled to 0 °C. Acetic anhydride (1.3 mL, ~13 mmol) was added slowly while stirring. The solution was allowed to reach RT overnight, followed by addition of saturated NH₄Cl and extraction with DCM (3 × 50 mL). The organic phases were combined and dried with MgSO₄ and concentrated under

reduced pressure to afford a pale-yellow syrup (0.195 g, 0.735 mmol, 55%). ¹H NMR (500 MHz, DMSO-$d_6$, δ from residual protio solvent) δ 12.40 (s, br, 1H), 10.00 (s, 1H), 8.03 (s, 1H), 7.67 (d, J = 8.3 Hz, 1H), 7.23 (d, J = 8.3 Hz, 1H), 3.57 (s, 3H), 3.10 (t, J = 7.7 Hz, 2H), 2.56 (t, J = 7.7 Hz, 2H), and 2.03 (s, 3H). ¹³C NMR (126 MHz, DMSO, δ from solvent) δ 172.61, 168.32, 137.54, 135.83, 131.09, 130.43, 122.18, 120.75, 51.18, 35.08, 28.50, 23.88, and 20.99. LRMS-ESI (m/z): [M + H]⁺ Theoretical for C₁₃H₁₅NO₅: 266.1; Experimental: 266.1.

**methyl (R)−3-(2-((1-(naphthalen-1-yl)ethyl)carbamoyl)phenyl)propanoate (3).** A 20 mL DCM solution containing 2-(3-methoxy-3-oxopropyl)benzoic acid (0.500 g, 2.4 mmol) was cooled to 0 °C followed by addition of HBTU (1.138 g, 3.0 mmol). This solution was stirred for 30 min, followed by addition of (R)−1-(naphthalen-1-yl)ethan-1-amine (0.409 g, 2.4 mmol) and DIPEA (0.522 mL, 3.0 mmol). The solution was warmed to RT and stirred for 16 h. The reaction mixture was quenched with 50 mL of H₂O and extracted with DCM (3×50 mL). The organic layers were collected and dried with MgSO₄ and concentrated under reduced pressure. The residue was purified by silica gel chromatography using 3:1 Hexanes: EtOAc ($R_f$ = 0.36) to afford a white solid. Washes were performed, and the resulting solid was dried under reduced pressure. This workup afforded the product as an off-white solid (0.723 g, 2.0 mmol, 83%). ¹H NMR (500 MHz, DMSO-$d_6$) δ from residual protio solvent 8.95 (d, J = 7.9 Hz, 1H), 8.24 (d, J = 8.4 Hz, 1H), 7.95 (d, J = 8.0 Hz, 1H), 7.84 (d, J = 8.1 Hz, 1H), 7.65−7.46 (m, 4H), 7.38−7.29 (m, 2H), 7.30−7.23 (m, 2H), 5.92 (p, J = 7.2 Hz, 1H), 3.57 (s, 3H), 2.92 (t, J = 8.0 Hz, 2H), 2.57 (t, J = 7.9 Hz, 2H), 1.58 (d, J = 6.9 Hz, 3H). ¹³C NMR (126 MHz, DMSO, δ from solvent): 172.51, 168.02, 140.12, 138.11, 136.96, 133.36, 130.39, 129.56, 129.34, 128.62, 127.29, 127.19, 126.11, 126.00, 125.56, 125.43, 123.11, 122.46, 51.21, 44.36, 34.96, 27.96, and 21.36. HRMS-ESI (m/z): [M + H]⁺ Theoretical for C₂₃H₂₄NO₃: 362.1756; Experimental: 362.1745.

**methyl (R)-3-(4-acetamido-2-((1-(naphthalen-1-yl)ethyl)carbamoyl)phenyl)propanoate (4).** Compound **4** was prepared similarly to the amide coupling of **3**. The amount of materials used were: **2** (0.350 g, 1.08 mmol); HBTU (0.899 g, 2.15 mmol); (R)-1-(naphthalen-1-yl)ethan-1-amine (0.366 g, 2.15 mmol); and DIPEA (0.749 mL, 4.30 mmol). Silica gel column purification was performed under a gradient from 1:1, 2:1, 3:1 EtOAc:Hexanes at 1 column volume for each gradient step. Compound **4** was isolated as white solid (0.410 g, 0.980 mmol, 91%). ¹H NMR (500 MHz, DMSO-$d_6$, δ from residual protio solvent) δ 9.96 (s, 1H), 8.95 (d, J = 8.0 Hz, 1H), 8.24 (d, J = 8.4 Hz, 1H), 7.95 (dd, J = 8.0, 1.6 Hz, 1H), 7.84 (d, J = 8.2 Hz, 1H), 7.64−7.55 (m, 3H), 7.54 (ddd, J = 8.1, 6.8, 1.3 Hz, 1H), 7.52−7.45 (m, 2H), 7.17 (d, J = 8.4 Hz, 1H), 5.92 (p, J = 7.2 Hz, 1H), 3.56 (s, 3H), 2.83 (t, J = 7.8 Hz, 2H), 2.69 (s, 3H), 2.53 (t, J = 8.0 Hz, 2H), 2.02 (s, 3H), and 1.57 (d, J = 6.9 Hz, 3H). ¹³C NMR (126 MHz, DMSO, δ from solvent) δ 172.50, 168.22, 167.88, 140.07, 137.33, 137.26, 133.33, 132.26, 130.39, 129.78, 128.60, 127.19, 126.14, 125.56, 125.36, 123.08, 122.39, 119.69, 117.71, 51.17, 44.22, 38.19, 35.02, 27.39, 23.85, and 21.39. LRMS-ESI (m/z): [M + H]⁺ Theoretical for C₂₅H₂₆N₂O₄: 419.2; Experimental: 419.2.

**(R)-2-(3-hydrazineyl-3-oxopropyl)-N-(1-(naphthalen-1-yl)ethyl)benzamide (5).** To a 10 mL EtOH solution containing **1** (0.400 g, 1.11 mmol) was added 0.5 mL (~1 M) of hydrazine monohydrate (N₂H₄ 64−65%, reagent grade 95%). The pale-yellow, homogenous solution was refluxed for 16 h. The resulting solution was reduced under vacuum to afford an off-white powder. To remove excess hydrazine monohydrate, several (3 × 15 mL) Et₂O washes were performed, and the resulting solid was dried under reduced pressure. This workup afforded the product as an off-white solid (0.390 g, 1.08 mmol, 97%). ¹H NMR (500 MHz, DMSO-$d_6$, δ from residual protio solvent): 8.97 (d, J = 7.9 Hz, 1H), 8.91 (s, 1H), 8.25 (d, J = 8.5 Hz, 1H), 7.96 (d, J = 8.1 Hz, 1H), 7.84 (d, J = 8.1 Hz, 1H), 7.65 (d, J = 7.2 Hz, 1H), 7.61 (t, J = 7.6 Hz, 1H),

7.54 (dt, $J = 15.0$, 7.6 Hz, 2H), 7.35 (t, $J = 7.4$ Hz, 1H), 7.31 (d, $J = 7.4$ Hz, 1H), 7.28–7.21 (br, 2H), 5.93 (p, $J = 7.2$ Hz, 1H), 4.21 (s, 2H), 2.91 (td, $J = 7.5$, 4.3 Hz, 2H), 2.35 (t, $J = 7.9$ Hz, 2H), and 1.60 (d, $J = 6.9$ Hz, 3H). $^{13}$C NMR (126 MHz, DMSO, δ from solvent): 170.82, 168.04, 140.20, 138.74, 137.05, 133.35, 130.37, 129.22, 129.20, 128.61, 127.21, 127.16, 126.14, 125.72, 125.55, 125.50, 123.12, 122.46, 44.42, 34.85, 28.22, and 21.44. HRMS-ESI ($m/z$): $[M + H]^+$ Theoretical for $C_{22}H_{24}N_3O_2$: 362.1859; Experimental: 362.1885.

**(R)-5-acetamido-2-(3-hydrazinyl-3-oxopropyl)-N-(1-(naphthalen-1-yl)ethyl)benzamide (6).** Compound **6** was prepared analogously to **5**. The amounts of materials used were: **4** (0.400 g, 0.956 mmol); 10 mL EtOH solution containing; 0.5 mL (~1 M) of hydrazine monohydrate ($N_2H_4$ 64-65%, reagent grade 95%). This procedure afforded an off-white solid (0.388 g, 0.927 mmol, 97%) ($^1$H NMR (500 MHz, DMSO-$d_6$, δ from residual protio solvent) δ 9.94 (s, 1H), 8.97 (d, $J = 7.9$ Hz, 1H), 8.89 (s, 1H), 8.24 (d, $J = 8.4$ Hz, 1H), 7.95 (d, $J = 8.1$ Hz, 1H), 7.84 (d, $J = 8.0$ Hz, 1H), 7.65–7.56 (m, 3H), 7.53 (dt, $J = 18.1$, 7.5 Hz, 2H), 7.45 (s, 1H), 7.15 (d, $J = 8.4$ Hz, 1H), 5.92 (p, $J = 6.9$ Hz, 1H), 4.11 (s, br, 2H), 2.82 (hept, $J = 7.5$, 7.0 Hz, 2H), 2.31 (t, 2H), 2.01 (s, 3H), and 1.58 (d, $J = 7.0$ Hz, 3H). $^{13}$C NMR (126 MHz, DMSO, δ from solvent) δ 170.85, 168.18, 167.90, 140.17, 137.40, 137.02, 133.34, 132.89, 130.39, 129.42, 128.60, 127.18, 126.17, 125.57, 125.44, 123.11, 122.39, 119.68, 117.66, 44.31, 34.89, 27.65, 23.85, and 21.48. LRMS-ESI ($m/z$): $[M + H]^+$ Theoretical for $C_{25}H_{26}N_4O_3$: 419.2; Experimental: 419.2.

**Preparation of compounds with electrophilic warheads.** Compounds **7**, **9**, **11**, and **13** were prepared by taking 0.030 g (0.083 mmol) of **5** and 0.029 mL (0.166 mmol) of DIPEA into 5 mL anhydrous DCM under N$_2$ atmosphere. Once dissolved, 0.100 mmol (1.2 equiv.) of appropriate acid chloride was added while stirring under N$_2$ atmosphere. Rapid reaction resulted in precipitation of a white solid. The reaction was left at RT for 2 h with no observable changes. The DCM was removed under reduced pressure and Et$_2$O was added to the remaining residue to precipitate a white solid that was collected with a 2 mL fritted glass funnel. The remaining white solid was washed extensively with Et$_2$O, dried, and collected. Isolated yields: **7** (0.022 g, 0.046 mmol, 56%); **9** (0.018 g, 0.041 mmol, 50%); **11** (0.020 g, 0.047 mmol, 56%); and **13** (0.024 g, 0.050 mmol, 60%).

Compounds **8** and **10** were prepared by placing 0.040 g (0.096 mmol) of **6** in 5 mL of anhydrous DMF followed by addition of K$_2$CO$_3$ (0.020 g, 0.145 mmol). The solution was stirred while 0.115 mmol (1.2 equiv.) of appropriate acid chloride was added. The solution was stirred at RT for 2 h followed by addition of 25 mL EtOAc and extraction with 3 × 25 mL of H$_2$O to remove DMF. The organic layers were combined, dried with MgSO$_4$, and concentrated under reduced pressure. The crude residue was purified by silica gel flash chromatography using pure EtOAc with 1–5% MeOH to yield white solids: **8** (0.016 g, 0.032 mmol, 34%); **10** (0.019 g, 0.036 mmol, 37%).

**methyl-(R,E)-4-(2-(3-(2-((1-(naphthalen-1-yl)ethyl)carbamoyl)phenyl)propanoyl)hydrazinyl)-4-oxobut-2-enoate (7).** $^1$H NMR (500 MHz, DMSO-$d_6$, δ from residual protio solvent) δ 10.53 (s, 1H), 10.16 (s, 1H), 8.93 (d, $J = 7.9$ Hz, 1H), 8.24 (d, $J = 8.6$ Hz, 1H), 7.95 (d, $J = 8.1$ Hz, 1H), 7.83 (d, $J = 8.2$ Hz, 1H), 7.67–7.57 (m, 2H), 7.56–7.48 (m, 2H), 7.39–7.21 (m, 4H), 7.08 (d, $J = 15.6$ Hz, 1H), 6.69 (dd, $J = 15.5$, 1.9 Hz, 1H), 5.93 (p, $J = 7.3$ Hz, 1H), 3.75 (s, 3H), 2.94 (dt, $J = 8.8$, 5.0 Hz, 2H), 2.55–2.47 (m, 3H *overlaps with DMSO-$d_6$*), and 1.59 (d, $J = 6.9$ Hz, 3H). $^{13}$C NMR (126 MHz, DMSO, δ from solvent) δ 170.30, 168.56, 165.73, 161.57, 140.66, 139.06, 137.56, 135.58, 133.86, 130.91, 129.90, 129.82, 129.77, 129.13, 127.73, 127.70, 126.67, 126.35, 126.08, 126.02, 123.65, 122.96, 52.59, 44.94, 35.14, 28.56, and 21.92. HRMS-ESI ($m/z$): $[M + H]^+$ Theoretical for $C_{27}H_{28}N_3O_5$: 474.2029; Experimental: 474.2007.

**methyl-(R,E)-4-(2-(3-(4-acetamido-2-((1-(naphthalen-1-yl)ethyl)carbamoyl)phenyl)propanoyl)hydrazinyl)-4-oxobut-2-enoate (8).** $^1$H NMR (500 MHz, DMSO-$d_6$, δ from residual protio solvent) δ 10.52 (s, 1H), 10.15 (s, 1H), 9.95 (s, 1H), 8.93 (d, $J = 7.9$ Hz, 1H), 8.24 (d, $J = 8.6$ Hz, 1H), 7.95 (d, $J = 8.0$ Hz, 1H), 7.83 (d, $J = 8.2$ Hz, 1H), 7.64–7.57 (m, 3H, 7.56–7.48 (m, 2H), 7.45 (s, 1H), 7.21 (d, $J = 8.4$ Hz, 1H), 7.07 (d, $J = 15.6$ Hz, 1H), 6.68 (d, $J = 15.5$ Hz, 1H), 5.93 (p, $J = 7.2$ Hz, 1H), 3.75 (s, 3H), 2.86 (m, 2H), 2.47 (m, 2H), 2.02 (s, 3H), and 1.57 (d, $J = 6.9$ Hz, 3H). $^{13}$C NMR (126 MHz, DMSO, δ from solvent) δ 169.79, 168.17, 167.88, 165.18, 161.03, 140.09, 137.38, 137.09, 135.02, 133.31, 132.65, 130.37, 129.57, 129.22, 128.57, 127.17, 126.16, 125.55, 125.41, 123.09, 122.34, 119.68, 117.60, 52.04, 44.27, 34.64, 27.45, 23.83, and 21.41. HRMS-ESI ($m/z$): $[M + H]^+$ Theoretical for $C_{29}H_{31}N_4O_6$: 531.2244; Experimental: 531.2217.

**(R)-2-(3-(2-(2-chloroacetyl)hydrazinyl)-3-oxopropyl)-N-(1-(naphthalen-1-yl)ethyl)benzamide (9).** $^1$H NMR (500 MHz, DMSO-$d_6$, δ from residual protio solvent) δ 10.21 (s, 1H), 9.98 (s, 1H), 8.95 (d, $J = 7.8$ Hz, 1H), 8.24 (d, $J = 8.5$ Hz, 1H), 7.96 (d, $J = 8.1$ Hz, 1H), 7.84 (d, $J = 8.1$ Hz, 1H), 7.67–7.49 (m, 4H), 7.38–7.23 (m, 4H), 5.93 (p, $J = 7.2$ Hz, 1H), 4.14 (s, 2H), 2.94 (t, $J = 9.1$, 2H), 2.48 (t, $J = 9.1$ Hz, 2H), and 1.60 (d, $J = 6.8$ Hz, 3H). $^{13}$C NMR (126 MHz, DMSO, δ from solvent) δ 170.08, 168.06, 164.65, 140.15, 138.56, 137.04, 133.35, 130.39, 129.39, 129.31, 128.62, 127.20 (two overlapping $^{13}$C signals), 126.16, 125.83, 125.58, 125.51, 123.14, 122.45, 44.43, 40.86, 34.62, 28.02, 21.41 HRMS-ESI ($m/z$): $[M + H]^+$ Theoretical for $C_{24}H_{25}ClN_3O_3$: 438.1584; Experimental: 438.1565.

**(R)-5-acetamido-2-(3-(2-(2-chloroacetyl)hydrazinyl)-3-oxopropyl)-N-(1-(naphthalen-1-yl)ethyl)benzamide (10).** $^1$H NMR (500 MHz, DMSO-$d_6$, δ from residual protio solvent) δ 10.20 (s, 1H), 9.96 (s, 2H), 8.94 (d, $J = 8.0$ Hz, 1H), 8.25 (d, $J = 8.5$ Hz, 1H), 7.96 (d, $J = 8.1$ Hz, 1H), 7.85 (d, $J = 8.2$ Hz, 1H), 7.62 (q, $J = 6.7$ Hz, 3H), 7.54 (m, 2H), 7.46 (s, 1H), 7.21 (d, $J = 8.4$ Hz, 1H), 5.93 (q, $J = 7.3$ Hz, 1H), 4.14 (s, 2H), 2.86 (m, 2H), 2.45 (t, $J = 7.9$ Hz, 2H), 2.03 (s, 3H), and 1.58 (d, $J = 6.8$ Hz, 3H). $^{13}$C NMR (126 MHz, DMSO, δ from solvent) δ 170.60, 168.72, 168.43, 165.14, 140.64, 137.92, 137.63, 133.86, 133.21, 130.92, 130.12, 129.12, 127.72, 126.71, 126.10, 125.97, 123.64, 122.89, 120.23, 118.13, 44.82, 41.37, 35.18, 27.97, 24.38, and 21.96. HRMS-ESI ($m/z$): $[M + H]^+$ Theoretical for $C_{26}H_{28}ClN_4O_4$: 495.1799; Experimental: 495.1788.

**(R)-2-(3-(2-(2-cyanoacetyl)hydrazinyl)-3-oxopropyl)-N-(1-(naphthalen-1-yl)ethyl)benzamide (11).** $^1$H NMR (500 MHz, DMSO-$d_6$, δ from residual protio solvent) δ 10.16 (s, 1H), 9.96 (s, 1H), 8.93 (d, $J = 7.8$ Hz, 1H), 8.23 (d, $J = 8.5$ Hz, 1H), 7.95 (d, $J = 8.2$ Hz, 1H), 7.83 (d, $J = 8.2$ Hz, 1H), 7.66–7.57 (m, 2H), 7.57–7.48 (m, 2H), 7.39–7.21 (m, 4H), 5.92 (p, $J = 7.1$ Hz, 1H), 3.74 (s, 2H), 2.97–2.89 (t, 7.6 Hz, 2H), 2.47 (t, $J = 7.6$ Hz, 2H), and 1.59 (d, $J = 6.9$ Hz, 3H). $^{13}$C NMR (126 MHz, DMSO, δ from solvent) δ 170.13, 168.03, 161.12, 140.14, 138.52, 137.02, 133.34, 130.39, 129.39, 129.29, 128.61, 127.20, 127.18, 126.15, 125.82, 125.57, 125.50, 123.13, 122.44, 115.62, 44.41, 34.55, 27.99, 23.67, and 21.39. HRMS-ESI ($m/z$): $[M + H]^+$ Theoretical for $C_{25}H_{25}N_4O_3$: 429.1928; Experimental: 429.1949.

**(R)-N-(1-(naphthalen-1-yl)ethyl)-2-(3-oxo-3-(2-propioloylhydrazineyl)propyl)benzamide (12).** Compound **12** was synthesized under the same conditions as compounds **7**, **9**, **11**, and **13** except the initial coupling to the hydrazide of **5** was achieved with 3-(trimethylsilyl)propioloyl chloride. The DCM was removed under reduced pressure and the crude material was immediately dissolved in 1:1 THF:MeOH (6 mL total volume) and 10 mg of K$_2$CO$_3$ was added. The solution was stirred and monitored by TLC until the reaction was complete, approximately 30 min. The solution was concentrated and purified by silica gel flash chromatography (2:1 EtOAc:Hexanes) to yield 8 mg (0.019 mmol, 23%) of a pale yellow solid. $^1$H NMR (500 MHz,

Acetone-$d_6$, δ from residual protio solvent) δ 9.49 (s, 1H), 9.12 (s, 1H), 8.33 (d, $J$ = 8.6 Hz, 1H), 8.05 (d, $J$ = 8.2 Hz, 1H), 7.93 (d, $J$ = 8.2 Hz, 1H), 7.83 (d, $J$ = 8.2 Hz, 1H), 7.72 (d, $J$ = 7.2 Hz, 1H), 7.63 (t, $J$ = 7.8 Hz, 1H), 7.52 (m, 2H), 7.40–7.28 (m, 3H), 7.19 (m, 1H), 6.12 (p, $J$ = 7.3 Hz, 1H), 3.14–3.00 (m, $J$ = 7.4 Hz, 2H), 2.79 (s, 1H), 2.64 (m, 2H), 1.74 (d, $J$ = 6.8 Hz, 3H). ¹³C NMR (126 MHz, Acetone, δ from solvent) δ 171.42, 169.30, 151.76, 140.76, 140.01, 138.07, 134.97, 132.12, 130.72, 130.42, 129.63, 128.52, 128.23, 127.15, 126.83, 126.52, 126.35, 124.37, 123.62, 79.90, 77.04, 76.70, 45.67, 36.17, and 21.58. HRMS-ESI ($m/z$): [M + H]⁺ Theoretical for $C_{25}H_{24}N_3O_3$: 414.1819; Experimental: 414.1852.

**(R)-2-(3-(2-(2-cyano-3-cyclopropylacryloyl)hydrazineyl)-3-oxopropyl)-N-(1-(naphthalen-1-yl)ethyl)benzamide (13).** ¹H NMR (500 MHz, Acetone-$d_6$, δ from residual protio solvent) δ 8.32 (d, $J$ = 8.6 Hz, 1H), 7.99–7.91 (m, 2H), 7.83 (d, $J$ = 8.3 Hz, 1H), 7.71–7.68 (m, 2H), 7.63–7.58 (m, 1H), 7.57–7.46 (m, 2H), 7.39 (m, 1H), 7.32 (m, 2H), 7.21 (t, $J$ = 7.1 Hz, 1H), 6.10 (p, $J$ = 7.5 Hz, 1H), [1:2.5 $E$:$Z$ isomer ratio; 4.51 (dd, $J$ = 25.6, 7.6 Hz); 4.24 (dd, $J$ = 54.2, 11.8 Hz, 1H)], 3.21–2.98 (m, 4H), 2.77 (s, 1H), 1.73 (d, $J$ = 6.9 Hz, 3H), 1.18–1.02 (m, 1H), 0.70–0.56 (m, 2H), and 0.56–0.41 (m, 2H). Many multiple peaks with close δ spacings were observed in the ¹³C NMR presumably due to the $E$:$Z$ isomer mixture, these values are reported as observed. ¹³C NMR (126 MHz, Acetone-$d_6$, δ from solvent) δ 169.39, 169.21, 169.17, 169.15, 169.11, 169.06, 166.20, 166.09, 140.81, 140.80, 140.06, 140.05, 140.03, 139.99, 138.12, 138.09, 134.96, 132.10, 132.08, 131.02, 131.00, 130.47, 130.45, 129.68, 129.65, 128.53, 128.51, 128.50, 128.17, 128.14, 127.12, 127.11, 126.91, 126.51, 126.37, 126.32, 124.33, 124.31, 123.53, 123.49, 123.47, 115.74, 115.72, 114.81, 114.79, 64.23, 59.93, 45.69, 45.65, 45.62, 43.16, 43.13, 43.02, 42.97, 39.15, 39.11, 39.07, 30.30, 30.15, 29.99, 29.84, 29.69, 29.53, 29.38, 29.10, 28.56, 28.54, 21.64, 21.60, 12.09, 12.05, 12.02, 3.47, 3.23, 3.20, 2.90, 2.13, and 2.10. HRMS-ESI ($m/z$): [M + H]⁺ Theoretical for $C_{29}H_{29}N_4O_3$: 481.2240; Experimental: 481.2289.

**Preparation of noncovalent derivatives of GRL0617.** Compounds **14** and **15** were prepared analogously to the amide coupling of **3**. The amount of materials used were: 2-methylbenzoic acid (0.250 g, 1.80 mmol); 5-acetamido-2-methylbenzoic acid (0.348 g, 1.80 mmol); HBTU (0.853 g, 2.25 mmol); (R)-1-(naphthalen-1-yl)ethan-1-amine (0.306 g, 1.80 mmol); and DIPEA (0.392 mL, 2.25 mmol). Silica gel column purification was performed on **14** (3:1 Hexanes:EtOAc) and **15** (5% MeOH in DCM) to yield white solids **14** (0.463 g, 1.61 mmol, 89%); **15** (0.519 g, 1.50 mmol, 83%).

**(R)-2-methyl-N-(1-(naphthalen-1-yl)ethyl)benzamide (14).** ¹H NMR (500 MHz, DMSO-$d_6$, δ from residual protio solvent) δ 8.86 (d, $J$ = 8.0 Hz, 1H), 8.25 (d, $J$ = 8.4 Hz, 1H), 7.96 (d, $J$ = 8.0 Hz, 1H), 7.85 (d, $J$ = 8.1 Hz, 1H), 7.66–7.49 (m, 4H), 7.35–7.28 (m, 2H), 7.25–7.19 (m, 2H), 5.93 (p, $J$ = 7.2 Hz, 1H), 2.30 (s, 3H), and 1.59 (d, $J$ = 6.9 Hz, 3H). ¹³C NMR (126 MHz, DMSO, δ from solvent)) δ 168.09, 140.25, 137.22, 135.01, 133.35, 130.40, 130.23, 129.07, 128.62, 127.18, 126.96, 126.08, 125.55, 125.43, 125.36, 123.17, 122.49, 44.26, 21.42, and 19.21. HRMS-ESI ($m/z$): [M + H]⁺ Theoretical for $C_{20}H_{20}NO$: 290.1545; Experimental: 290.1594.

**(R)-5-acetamido-2-methyl-N-(1-(naphthalen-1-yl)ethyl)benzamide (15).** ¹H NMR (500 MHz, DMSO-$d_6$, δ from residual protio solvent) δ 9.91 (s, 1H), 8.85 (d, $J$ = 8.1 Hz, 1H), 8.24 (d, $J$ = 8.5 Hz, 1H), 7.96 (dd, $J$ = 8.1, 1.6 Hz, 1H), 7.84 (d, $J$ = 8.2 Hz, 1H), 7.64–7.45 (m, 7H), 7.12 (d, $J$ = 8.3 Hz, 1H), 5.92 (p, $J$ = 7.1 Hz, 1H), 3.29 (s, 1H), 2.69 (s with broadened couplings, 3H), 2.21 (s, 3H), 2.01 (d, $J$ = 1.7 Hz, 3H), 1.57 (d, $J$ = 6.9 Hz, 3H), and 1.19 (s, 1H). ¹³C NMR (126 MHz, DMSO, δ from solvent) δ 168.12, 167.96, 140.20, 137.52, 136.78, 133.33, 130.40, 130.38, 129.10, 128.59, 127.18, 126.10, 125.55, 125.37, 123.15, 122.42, 119.51, 117.50, 44.15, 38.19, 23.84, 21.44, and 18.51. HRMS-ESI ($m/z$): [M + H]⁺ Theoretical for $C_{22}H_{23}N_2O_2$: 369.1579; Experimental: 369.1555.

## Protein expression and purification

PLpro from SARS-CoV-2 was produced using a previously described procedure with minor modifications[62], which we summarize here. First, the protein was expressed using *E. coli* BL21(DE3) cells that had been transformed with a pMCSG92 expression plasmid, which includes a T7 promoter and TEV protease-cleavable C-terminal 6xHis tag. Cells were plated on LB agar and cultivated in a shaking incubator (250 rpm) at 37 °C in Lysogeny Broth medium (Lennox recipe) using 1 L per baffled 2.8 L Fernbach flask. Carbenicillin was used for antibiotic selection throughout. Bacterial growth was monitored by measuring the absorbance at 600 nm ($OD_{600}$). Upon reaching an $OD_{600}$ of ~0.7, the incubator temperature was set to 18 °C and isopropyl β-ᴅ-1-thiogalactopyranoside (IPTG) was added to 0.2 mM. After approximately 18 hours, the culture was harvested by centrifugation at 6000×$g$ for 30 min. After decanting off the supernatant, the pellets were stored at −80 °C until needed for protein purification.

A cell pellet harvested from a 1 L culture was thawed and resuspended in 100 mL of lysis buffer containing 50 mM HEPES, 300 mM NaCl, 50 mM imidazole, 5% glycerol, and 1 mM TCEP at pH 7.4. Following resuspension, the cells were subjected to tip sonication on ice at 50% amplitude (2 s on and 10 s off) for a total sonication time of 5 min using a Branson 450D Digital Sonifier. After clarifying the lysate by 38,500×$g$ centrifugation for 35 min at 4 °C, the decanted supernatant was passed through 1.6- and 0.45-micron syringe filters sequentially and kept on ice while loading a 5-mL HisTrap HP column (Cytiva) at 2 mL/min. After washing the column with 10 column volumes (CV) of lysis buffer, partially purified PLpro was eluted using a linear gradient (20 CVs) of lysis buffer with 500 mM imidazole. Elution fractions (2 mL) were collected and PLpro was identified using SDS-PAGE on a 4–20% Mini-Protean TGX Stain-Free protein gel (Bio-Rad). Pooled fractions containing PLpro were dialyzed overnight at 6 °C in 50 mM HEPES pH 7.4 with 150 mM NaCl, 5% glycerol, 20 mM imidazole, and 1 mM TCEP in the presence of His-tagged TEV protease (1 mg TEV protease:100 mg PLpro). After confirming His-tag cleavage by SDS-PAGE, the dialyzed protein solution was passed over a 5-mL HisTrap HP column to remove His-tagged impurities. The column flowthrough was collected, evaluated with SDS-PAGE, and concentrated with a 10-kDa molecular weight cutoff Amicon Ultra15 ultrafiltration membrane. Upon concentration, partially purified protein was applied at 0.5 mL/ min to a Superdex 75 10/300 GL size-exclusion column (Cytiva) that had been equilibrated with 50 mM Tris HEPES pH 7.4 with 150 mM NaCl, 5% glycerol, and 1 mM TCEP. Fractions (0.5 mL) containing purified PLpro were collected, pooled, and concentrated for further use.

## PLpro inhibition assays

The assays were performed in 40 µL total volume in black half area 96-well plates (Greiner PN 675076) at 25 °C. The assay buffer contained 20 mM Tris-HCl pH 7.45, 0.1 mg/mL bovine serum albumin fraction V, and 2 mM reduced glutathione. The final DMSO concentration in all assays was 2.5% v/v. PLpro initial rates were measured using a fluorogenic peptide substrate assay[15,21,22]. The substrates Z-LRGG-AMC and Z-RLRGG-AMC were purchased from Bachem (PN 4027157 and 4027158), dissolved to 10 mM in DMSO and stored in aliquots at −20 °C. To determine Michaelis−Menten parameters, 20 µL enzyme solution was dispensed into wells (250 nM final concentration), and reactions were initiated by adding 20 µL substrate to 0−500 µM final concentration, in duplicate. Release of aminomethylcoumarin (AMC) was monitored by a Biotek Synergy H1 fluorescence plate reader every 50 s with an excitation wavelength of 345 nm and an emission wavelength of 445 nm, 6.25 mm read height, and gain = 60. After background subtraction of the average of no-enzyme negative controls, product formation was quantified using a 0.02−5 µM calibration curve of AMC (Sigma PN 257370). Initial rates were determined for time points in the initial linear range by linear regression in Excel, and GraphPad Prism 9

was used to perform nonlinear regression of the Michaelis-Menten equation to the initial rate vs. substrate concentration data to yield $K_M$ and $V_{max}$.

Inhibitors were characterized by dispensing 10 μL enzyme solution into wells (115 nM final concentration), followed by 10 μL inhibitor solution at 4X desired final concentrations in 5% v/v DMSO in duplicate, centrifuging briefly, and incubating for 30 min. Reactions were initiated by adding 20 μL substrate to 100 μM final concentration. Initial rates were determined as described above and % residual activities were determined by normalizing to the average of no inhibitor controls (100% activity). Thirty-minute $IC_{50}$ values were determined by nonlinear regression to the [Inhibitor] vs. normalized response – Variable slope equation using GraphPad Prism 9.

Time-dependent inhibition assays were performed as described above, except that preincubation times were varied by adding the inhibitor to the enzyme at specific time points. For each inhibitor concentration, initial rates were normalized such that 0 preincubation time is 100% and plotted against preincubation time. A nonlinear regression to a one-phase decay model was performed to determine the rate constants $k_{obs}$ for each concentration and their 95% confidence intervals. These rate constants were then plotted against inhibitor concentration, and the data in the initial linear region was fit to determine the slope, which is $k_{inact}/K_I$. All regressions were performed with GraphPad Prism 9. We note that $k_{inact}/K_I$ is only valid when the testing concentrations are at least 10-fold below $K_i$, so there may be inaccuracies when this condition is not met.

### Inhibition of full-length Nsp3 de-ISG15ylase activities
HEK293T cells were obtained from ATCC and were grown in 10 cm dishes and transiently transfected with pEF-HA-Nsp3 or pEF empty vector using lipofectamine 3000 (ThermoFisher). 24 hrs after transfection, cells were harvested and lysed in 1% NP-40 lysis buffer (50 mM Tris-HCl, pH 7.5, 150 mM NaCl, 10% glycerol, 1% NP-40, 1 mM phenylmethylsulfonyl fluoride (PMSF)). Full-length HA-Nsp3 was purified using anti-HA immunoprecipitation (5 mg anti-HA antibody to 1 mg cell lysate), washed 4 times using the lysis buffer and the Nsp3-containing beads (~100 μl bead volume) were resuspended in 1.0 ml enzyme assay buffer (20 mM Tris-HCl, pH 8.0, 0.05% CHAPS, 2 mM β-mercaptoethanol). In all, 20 μl of the immunoprecipitated Nsp3 beads and the whole cell lysates (30 μg) were run on 8% SDS-PAGE, transferred to a polyvinylidene difluoride (PVDF) membrane, and probed with anti-HA antibody (1:1000 dilution) to detect full-length Nsp3. Activity of Nsp3 on the bead (5.0 μl) was monitored using ISG15-CHOP2 substrate (20 nM) in the presence of DMSO as vehicle or dose range of compounds in DMSO. After HA pulldown, Nsp3 activity assays were performed in 384-well plates. For each compound, the assay was performed in triplicate of dose responses. The assays were repeated two times (transfection, pulldown, and assay). Percent inhibition was calculated using the formula,

$$\% \text{ Inhibition} = 100 \times [1 - (X - \text{LOW})/(\text{HIGH} - \text{LOW})] \quad (1)$$

where X is the signal at a given concentration of inhibitor, LOW is the signal with no DUB added (100% inhibition) and HIGH is the signal with DUB in the presence of DMSO (0% inhibition). Percent inhibition was plotted using GraphPad Prism 9 and $IC_{50}$ values were determined using nonlinear regression to the [Inhibitor] vs. normalized response – Variable slope equation using GraphPad Prism 9.

### Mass spectrometry to assess covalent adduct formation
A Waters Synapt HDMS QTOF mass spectrometer was used to measure the intact protein mass of PLpro with and without preincubation with inhibitors to detect covalent adduct formation. To prepare the samples, 2 μL of 20 mM inhibitor stocks in DMSO were added to 100 μL PLpro at 1 mg/mL concentration and incubated 1 h at room temperature. Previously described protocols for ultrafiltration and denaturing direct infusion[63] were implemented as follows. Samples were processed by ultrafiltration with a Vivaspin 500 10 kDa PES membrane by diluting the sample to 0.5 mL with 10 mM LC-MS grade ammonium acetate and reducing volume to 50 μL twice, followed by the same procedure with 2.5 mM ammonium acetate. Protein concentrations were estimated by A280 with a NanoDrop 2000, and samples were diluted to 2 mg/mL in 2.5 mM ammonium acetate, and then 10 μL were further diluted into 90 μL 50:50 acetonitrile:water with 0.1% formic acid. Sample was introduced into the electrospray ionization source by syringe pump at a flow rate of 10 μL/min and MS1 spectra were collected for m/z 400–1500, 5 s/scan, for 1 min. The protein monoisotopic mass was determined from the averaged spectra using mMass 5.5[64].

### Inhibition of PLpro deubiquitinase and de-ISG15ylase activities and deubiquitinase selectivity
Candidate inhibitors were assayed by LifeSensors, Inc. (Malvern, PA) in quadruplicate for inhibition of SARS-CoV-2 PLpro with Ub-rhodamine or ISG15-CHOP2 and with human deubiquitinase (DUB) enzymes, including USP30, USP15, USP8, USP7, USP4, and USP2C as well as UCH-L1 with Ub-rhodamine, except for USP7, which was tested with Ub-CHOP2. The CHOP assay[65] uses a quenched enzyme platform to quantify the DUB inhibition activity of the compounds. In this assay, a reporter enzyme is fused to the C-terminus of ubiquitin. The reporter is silent when fused to ubiquitin but becomes fluorescent upon cleavage from the C-terminus by a DUB. Thus, measurement of the reporter activity is a direct measure of DUB activity. Assays were performed with a positive control (PR619) and negative control (i.e., without the inhibitor). DUBs at previously optimized concentrations were used with previously optimized suitable DUB substrates to evaluate inhibitory activity. Briefly, the received compounds in DMSO were thawed before use and simultaneously aliquoted to protect against deterioration from freeze-thaw cycles. Compounds were diluted at desired fold to measure a dose response curve in DMSO. DMSO control was used as 0% inhibition in the presence of DUB and the DMSO control without the DUB was considered as the 100% inhibition control to calculate $IC_{50}$ values. Dose response-inhibition curves were plotted in GraphPad Prism with log-transformed concentration on the X-axis with percentage inhibition (30 min time point) on the Y-axis using log [inhibitor] versus the response-variable slope. The selectivity index (SI) is the fold change in selectivity for PLpro compared to the DUB inhibition activity of other DUBs in the selectivity panel.

### PLpro expression, purification, and crystallization
Wild-type PLpro from SARS-CoV-2 was expressed in BL21(DE3) *E. coli* cells transformed with the pMCSG53 expression plasmid with a T7 promoter and a TEV-cleavable, N-terminal 6xHis-tagged PLpro. *E. coli* cells were grown in LB media containing 50 μg/mL ampicillin at 37 °C in a shaking incubator (200 rpm) until the optical density ($OD_{600}$) of the culture was 0.6. The culture was then induced with 0.5 mM IPTG (GoldBio, USA) and grown for 16 h at 18 °C. The culture was centrifuged for 15 min at 3000×g and the cells were obtained as pellets. *E. coli* pellets were resuspended in lysis buffer (50 mM HEPES pH 7.2, 150 mM NaCl, 5% glycerol, 20 mM imidazole, 10 mM 2-mercaptoethanol) and subjected to sonication for cell lysis. The soluble fraction of the whole cell lysate was separated by centrifugation at 20442×g for 80 min and was loaded onto a Ni-NTA Agarose (Qiagen, USA) gravity column pre-equilibrated with lysis buffer. The column was washed with 25 column volumes of wash buffer (50 mM HEPES pH 7.2, 150 mM NaCl, 5% glycerol, 50 mM imidazole, 10 mM 2-mercaptoethanol) and eluted in fractions with elution buffer (50 mM HEPES pH 7.2, 150 mM NaCl, 5% glycerol, 500 mM imidazole, 10 mM 2-mercaptoethanol). Fractions containing PLpro protein as determined by SDS-PAGE were combined and dialyzed overnight in dialysis buffer (50 mM HEPES pH

7.2, 150 mM NaCl, 5% glycerol, 10 mM 2-mercaptoethanol). Dialyzed PLpro was mixed with 6xHis-tagged TEV protease in 25:1 ratio, incubated overnight at 4 °C and was passed through Ni-NTA Agarose (Qiagen, USA) gravity column pre-equilibrated with dialysis buffer (50 mM HEPES pH 7.2, 150 mM NaCl, 5% glycerol, 10 mM 2-mercaptoethanol) to remove 6xHis-tagged impurities and TEV protease. Tagless PLpro obtained as the flowthrough was flash frozen and stored at −80 °C. All extraction and purification steps were performed at 4 °C. Reaction of tagless PLpro in 20 mM Tris HCl pH 8.0 and 5 mM NaCl with a 10-fold molar excess of compound **7** was performed at 41 °C for 20 min. The PLpro-compound **7** complex in a solution containing 20 mM Tris HCl, 100 mM NaCl and 10 mM DTT was then used for crystallization at a concentration of 8 mg/ml. Initial crystal hits were obtained by screening around 900 crystallization conditions by the sitting drop method. Diffraction-quality crystals were obtained from a well solution containing PEG-3350, $CaCl_2$, $CdCl_2$, and $CoCl_3$.

### Data collection and structure determination

The diffraction data were collected at 100 K at the BL12-2 beamline of the Stanford Synchrotron Radiation Light Source using Pilatus 6 M detectors. Crystals for the complex were cryo-cooled using the well solution supplemented with 20% ethylene glycol. Diffraction data from two crystals were collected with 360° of data per crystal and 0.2° oscillation per image. For each crystal, diffraction data were merged and processed with the XDS suite of programs[66]. The structures were solved by molecular replacement with MOLREP[67] using the coordinates of SARS-CoV-2 PLpro complexed with the tetrapeptide-based inhibitor VIR251 (PDB 6WX4[11]) as the search model. Iterative rounds of model building and refinement were performed with the programs COOT[68] and REFMAC[69]. The details of data collection and refinement for the higher resolution data (3.10 Å) are presented in Supplementary Table 3.

### SARS-CoV-2 antiviral assays

Initial screening to measure cytopathic effect (CPE) protection for the 50% efficacy concentration ($EC_{50}$) and cytotoxicity ($CC_{50}$) was performed in the Regional Biocontainment Laboratory at the University of Tennessee Health Science Center using an assay based on African green monkey kidney epithelial (Vero E6) cells in 384-well plates[70]. Each plate can evaluate five compounds in duplicate at seven concentrations to measure an $EC_{50}$ and $CC_{50}$. Each plate included three controls: cells alone (uninfected control), cells with SARS-CoV-2 (infected control) for plate normalization, and remdesivir as a drug control. Cell viability was measured using the CellTiter-Glo Luminescent Cell Viability Assay (Promega). In brief, Vero E6 TMPRSS ACE2 cells (obtained from Dr. Barney Graham, NIH) were grown to ~90% confluency in 384-well plates and treated for 1 hr with compounds. Cells were infected at an MOI = 0.1 of SARS-CoV-2 isolate USA-WA1/2020[71]. After 48 h, the SARS-CoV-2-mediated CPE and cytotoxicity were assessed by measuring live cells using CellTiter-Glo. The selectivity index at 50% ($SI_{50}$) was then calculated from the $EC_{50}$ and $CC_{50}$ values. To ensure robust and reproducible signals, each 384-well plate was evaluated for its $Z$-score, signal to noise, signal to background, and coefficient of variation. This assay has been validated for use in high-throughput format for single-dose screening and is sensitive and robust, with $Z$ values > 0.5, signal to background >20, and signal to noise >3.3. Antiviral activity and cytotoxicity were also assessed with compound in the presence of 2 µM CP-100356 and SARS-CoV-2. Following incubation for 48 h at 5% $CO_2$ and 37 °C, the percent cell viability was measured with CellTiterGlo. Signals were read with an EnVision® 2105 multimode plate reader. Cells alone (positive control) and cells plus virus (negative control) were set to 100% and 0% cell viability to normalize the data from the compound testing. Data were normalized to cells (100%) and virus (0%) plus cells. Each concentration was tested in duplicate.

Compounds were also tested against SARS-CoV-2 variants using Vero E6 cells (obtained from ATCC) at the Institute for Antiviral Research at Utah State University under a service contract sponsored by NIAID using methods described previously[72]. Confluent or near-confluent cell culture monolayers of Vero E6 cells were prepared in 96-well disposable microplates the day before testing. Cells were maintained in Modified Eagle Medium (MEM) supplemented with 5% fetal bovine serum (FBS). For antiviral assays the same medium was used but with FBS reduced to 2% and supplemented with 50 µg/ml gentamicin. Compounds were dissolved in DMSO, saline, or the diluent requested by the submitter. Less soluble compounds were vortexed, heated, and sonicated, and if they still did not go into solution were tested as colloidal suspensions. Each test compound was prepared at four serial $log_{10}$ concentrations, usually 0.1, 1.0, 10, and 100 µg/ml or µM (per sponsor preference). Lower concentrations were used when insufficient compound was supplied. Five microwells were used per dilution: three for infected cultures and two for uninfected toxicity cultures. Controls for the experiment consisted of six microwells that were infected and not treated (virus controls) and six that were untreated and uninfected (cell controls) on every plate. A known active drug was tested in parallel as a positive control drug using the same method applied for test compounds. The positive control was tested with every test run.

Growth media was removed from the cells and the test compound was applied in 0.1 ml volume to wells at 2X concentration. Virus, normally at ~60 $CCID_{50}$ (50% cell culture infectious dose) in 0.1 ml volume, was added to the wells designated for virus infection. Medium devoid of virus was placed in toxicity control wells and cell control wells. Plates were incubated at 37 °C with 5% $CO_2$ until marked CPE (>80% CPE for most virus strains) was observed in virus control wells. The plates were then stained with 0.011% neutral red for approximately two hours at 37 °C in a 5% $CO_2$ incubator. The neutral red medium was removed by complete aspiration, and the cells were rinsed 1X with phosphate buffered saline (PBS) to remove residual dye. The PBS was removed completely, and the incorporated neutral red was eluted with 50% Sorensen's citrate buffer/50% ethanol for at least 30 min. Neutral red dye penetrates living cells. Thus, the more intense the red color, the larger the number of viable cells present in the wells. The dye content in each well was quantified using a spectrophotometer at 540 nm wavelength. The dye content in each set of wells was converted to a percentage of dye present in untreated control wells using a Microsoft Excel spreadsheet and normalized based on the virus control. The 50% effective $EC_{50}$ concentrations and 50% cytotoxic ($CC_{50}$) concentrations were then calculated by regression analysis. The quotient of $CC_{50}$ divided by $EC_{50}$ gives the selectivity index (SI). Compounds showing SI values ≥10 were considered active.

To confirm antiviral activity of compounds in human cells, we evaluated the compounds against SARS-CoV2 variants using a Caco-2 virus yield reduction assay. Caco-2 cells were obtained from ATCC. This test was performed at the Institute for Antiviral Research of Utah State University under a service contract sponsored by NIAID and following the method described previously[72]. Briefly, near-confluent monolayers of Caco-2 cells were prepared in 96-well microplates the day before testing. Cells were maintained in MEM supplemented with 5% FBS. The test compounds were prepared at a serial dilution of concentrations. The antiviral activity was also assessed with the compound alone or in the presence of 2 µM CP-100356. Three microwells were used per dilution. Controls for the experiment consisted of six microwells that were infected and not treated (virus controls) and six that were untreated and uninfected (cell controls) on every plate. A known active drug was tested in parallel as a positive control drug using the same method as is applied for test compounds. The positive control was tested with every test run. Growth media was removed from the cells and the test compound applied in 0.1 ml volume to wells at 2X concentration. Virus, normally at ~60 $CCID_{50}$ (50% cell culture

infectious dose) in 0.1 ml volume, was added to the wells designated for virus infection. Medium devoid of virus was placed in cell control wells. Plates were incubated at 37 °C with 5% $CO_2$. After sufficient virus replication occurs (3 days for SARS-CoV-2), a sample of supernatant was taken from each infected well (three replicate wells were pooled) and tested immediately for virus yield reduction (VYR) or held frozen at −80 °C for later virus titer determination.

The VYR test is a direct determination of how much the test compound inhibits virus replication. Virus yielded in the presence of test compound was titrated and compared to virus titers from the untreated virus controls. Titration of the viral samples (collected as described above) was performed by endpoint dilution. Serial 1/10 dilutions of virus were made and plated into four replicate wells containing fresh cell monolayers of Vero E6 cells. Plates were then incubated, and cells were scored for the presence or absence of virus after distinct CPE was observed, and the $CCID_{50}$ was calculated using the Reed–Muench method[58]. The 90% effective concentration ($EC_{90}$) was calculated by regression analysis by plotting the $log_{10}$ of the inhibitor concentration versus $log_{10}$ of virus produced at each concentration. $EC_{90}$ values were calculated from data to compare to the concentration of drug compounds as measured in the pharmacokinetic experiments. Drug concentrations in critical tissues above $EC_{90}$ values were targeted (instead of $EC_{50}$ values) as for clinically relevant applications.

## Metabolic stability
Intrinsic clearance in human, Sprague-Dawley rat, and CD-1 mouse liver microsomes and S9 fractions were measured[73] in duplicate for compounds **7**, **9**, and **14** by Eurofins Panlabs (St. Charles, MO, USA). Imipramine, propranolol, terfenadine, and verapamil were used as reference compounds at a test concentration of 0.1 μM. In each experiment and if applicable, the respective reference compounds were tested concurrently with the test compounds, and the data were compared with historical values determined at Eurofins. The experiments were accepted in accordance with Eurofins validation Standard Operating Procedure. Metabolic stability, expressed as percent of the parent compound remaining, was calculated by comparing the peak area of the compound at the time point relative to that at time $t_0$. The concentration of each compound was 1 μM and the incubation time ranged from 0 to 60 min. The half-life ($T_{1/2}$) was estimated from the slope of the initial linear range of the logarithmic curve of compound remaining (%) versus time, assuming first-order kinetics. The apparent intrinsic clearance ($CL_{int}$, μL/min/mg) was then calculated according to the following formula:

$$CL_{int} = \frac{0.693}{T_{1/2}(\text{mg protein/μL})} \quad (2)$$

## Pharmacokinetics
Compound **7** was formulated in 10% dimethyl sulfoxide (DMSO)/30% polyethylene glycol (PEG) 400/10% Kolliphor® EL/50% water for injection (WFI) at 1 and 0.6 mg/mL for PO and IV, respectively. A dosing volume of 10 mL/kg was applied for PO and 5 mL/kg for IV. Male ICR mice (age = 4–6 weeks) weighing 22 ± 2 g were provided by BioLasco Taiwan (under Charles River Laboratories Licensee). Animals were acclimated for three days prior to use and were confirmed with good health. All animals were maintained in a hygienic environment with controlled temperature (20–24 °C), humidity (30–70%) and 12-h light/dark cycles. Free access to sterilized standard lab diet (Oriental Yeast Co., Ltd., Japan) and autoclaved tap water were granted. In vivo PK experiments involved a total of 48 ICR (CD-1) mice separated into two groups of 24 mice each. One group was used to assess intravenous (i.v.) PK and the other group was used to assess oral (p.o.) PK. The sample sizes were chosen to allow three biological replicates at eight time

points for each group. The data were used to calculate mean values and standard error of the mean (SEM). Animals were acclimated for 3 days prior to use and were confirmed with good health. All animals were maintained in a hygienic environment with controlled temperature (20–24 °C), humidity (30–70%), and 12 h light/dark cycles. Free access to sterilized standard lab diet [MFG (Oriental Yeast Co., Ltd., Japan)] and autoclaved tap water were granted. Animals were euthanized by $CO_2$ for blood collection by cardiac puncture. Blood samples (300–400 μL) were collected in tubes coated with EDTA-K2, mixed gently, then kept on ice and centrifuged at 2500×$g$ for 15 min at 4 °C, within 1 h of collection. The plasma was then harvested and kept frozen at −70 °C until further processing.

The exposure levels (ng/mL) of **7** in plasma samples were determined by LC-MS/MS. Plots of plasma concentrations (mean ± SD) vs. time for **7** were constructed. The fundamental PK parameters after PO ($t_{1/2}$, $T_{max}$, $C_{max}$, $AUC_{last}$, $AUC_{Inf}$, $AUC/D$, $AUC_{extr}$, $MRT$, $V_z$, and $Cl$) and IV ($t_{1/2}$, $C_0$, $AUC_{last}$, $AUC_{Inf}$, $AUC/D$, $AUC_{extr}$, $MRT$, $V_{ss}$, and $Cl$) administrations were obtained from the noncompartmental analysis of the plasma data using WinNonlin (best-fit mode). The mean values of the data at each time point were used in the parameter analysis.

## Reporting summary
Further information on research design is available in the Nature Portfolio Reporting Summary linked to this article.

## Data availability
Structural data for the SARS-CoV-2 papain-like protease in complex with compound **7** were deposited in the Protein Data Bank (PDB) with accession code 8EUA. All other data generated or analyzed during this study are included in this published article (and its supplementary information files). Publicly available datasets used in this study are X-ray crystal structures of SARS-CoV-2 PLpro with accession codes PDB ID: 7JIR, 7CMD, 6WX4, 6W9C, 6WZU, 6XAA; a structure of UCH-L1 with PDB ID: 3KW5; and a structure of USP4 with PDB ID: 2Y6E. Source data are provided with this paper. Data are available from the corresponding authors upon request. Source data are provided with this paper.

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

## Acknowledgements

We thank Paul Abraham and Richard Giannone for assistance with mass spectrometry and Yunqiao (Joseph) Pu for assistance with nuclear magnetic resonance spectroscopy. We also thank Daniel Fernandez and the Macromolecular Structure Knowledge Center at Stanford for providing equipment for crystallography. We thank Bernd Meibohm for helpful discussions. This work was supported by the U.S. Department of Energy (DOE) Office of Science through the National Virtual Biotechnology Laboratory, a consortium of DOE national laboratories focused on response to COVID-19, with funding provided by the Coronavirus CARES Act. Use of the Stanford Synchrotron Radiation Lightsource, SLAC National Accelerator Laboratory, was supported by the U.S. Department of Energy (DOE), Office of Science, Office of Basic Energy Sciences under Contract No. DE-AC02-76SF00515. The SSRL Structural Molecular Biology Program is supported by the DOE Office of Biological and Environmental Research, and by the National Institutes of Health, National Institute of General Medical Sciences (P30GM133894). The contents of this publication are solely the responsibility of the authors and do not necessarily represent the official views of NIGMS or NIH. Funding for this project was provided in part by federal funds from the National Institute of Allergy and Infectious Diseases, National Institutes of Health, Department of Health and Human Services, under Contract HHSN272201700060C and 75N93022C00035 (A.J.). A.L. was supported in part by the Laboratory Directed Research and Development Program at Oak Ridge National Laboratory (ORNL) and through an appointment to the Science Education and Workforce Development Programs at ORNL, administered by Oak Ridge Institute for Science and Education (ORISE) through the U.S. DOE. Protein production was supported by the ORNL Center for Structural Molecular Biology funded by the DOE OBER. B.C.S. and J.M.P. also received funding from the Technology Innovation Program at ORNL. This research used resources at the Spallation Neutron Source, a U.S. Department of Energy Office of Science User Facility operated by ORNL, and at the Compute and Data Environment for Science (CADES) at ORNL, which is managed by UT Battelle, LLC, for DOE under contract DE-AC05–00OR22725.

## Author contributions

B.C.S., S.G., S.W., M.S.H., and J.M.P. conceived the study.

## Competing interests

B.C.S., S.G., and J.M.P. are inventors on invention disclosures and pending institutional patent applications on covalent PLpro inhibitors (17/896,182 and PCT/US2022/041629 filed 27 Aug 2022). B.C.S., L.F., and J.M.P. are also inventors on U.S. provisional application 63/454,205 filed 23 Mar 2023 on covalent PLpro inhibitors. The remaining authors declare no other competing interests.

## Additional information

[1]Biosciences Division, Oak Ridge National Laboratory, Oak Ridge, TN, USA. [2]Department of Chemical and Systems Biology, Stanford University School of Medicine, Stanford, CA, USA. [3]Biological Sciences Division, SLAC National Accelerator Laboratory, Menlo Park, CA, USA. [4]Stanford Synchrotron Radiation Lightsource, Menlo Park, CA, USA. [5]Neutron Scattering Division, Oak Ridge National Laboratory, Oak Ridge, TN, USA. [6]Department of Structural Biology, Stanford University School of Medicine, Stanford, CA, USA. [7]B-11 Bioenergy and Biome Sciences, Bioscience Division, Los Alamos National Laboratory, Los Alamos, NM, USA. [8]Department of Microbiology, Immunology and Biochemistry, University of Tennessee Health Science Center, Memphis, TN, USA. [9]Regional Biocontainment Laboratory, University of Tennessee Health Science Center, Memphis, TN, USA. [10]Institute for Antiviral Research, Department of Animal, Dairy, and Veterinary Sciences, Utah State University, Logan, UT, USA. [11]Progenra Inc., Malvern, PA, USA. [12]Biology Department, Brookhaven National Laboratory, Upton, NY, USA. [13]Center for BioMolecular Structure, National Synchrotron Light Source II, Brookhaven National Laboratory, Upton, NY, USA. [14]Center for Structural Genomics of Infectious Diseases, Consortium for Advanced Science and Engineering, University of Chicago, Chicago, IL, USA. [15]Biosciences Division, Argonne National Laboratory, Argonne, IL, USA. [16]Molecular Biosciences and Integrated Bioimaging, Lawrence Berkeley National Laboratory, Berkeley, CA, USA. [17]Department of Bioengineering, University of California, Berkeley, CA, USA. [18]Structural Biology Center, X-ray Science Division, Argonne National Laboratory, Argonne, IL, USA. [19]Department of Biochemistry and Molecular Biology, University of Chicago, Chicago, IL, USA. [20]Department of Chemistry and Chemical Biology, Northeastern University, Boston, MA, USA. [21]Joint Institute for Biological Sciences, Oak Ridge National Laboratory, Oak Ridge, TN, USA. [22]Computing and Computational Sciences Directorate, Oak Ridge National Laboratory, Oak Ridge, TN, USA. [23]Present address: Department of Process Research and Development, Merck & Co., Inc., Rahway, NJ, USA. [24]Present address: Computational and Data Sciences, Center for Research Acceleration by Digital Innovation, Amgen, Inc., Thosand Oaks, CA, USA. [25]These authors contributed equally: Brian C. Sanders, Suman Pokhrel. ✉e-mail: sandersbc@ornl.gov; soichi.wakatsuki@stanford.edu; parksjm@ornl.gov

