## [Peer Review File · Nature Communications]

Potent and Selective Covalent Inhibition of the Papain-like Protease from SARS-CoV-2REVIEWER COMMENTS

Reviewer #1 (Remarks to the Author):

In this manuscript, Sanders et al. describe GRL0617-based covalent inhibitors of the SARS-CoV-2 papain-like protease. This involved design and synthesis of a number of different analogs followed by extensive characterization of these molecules in a comprehensive combination of in vitro, cell-based, and animal studies. Overall, idea of using GRL0617 as a scaffold for design of a covalent PLpro inhibitor is clever and the study appears to be extremely rigorous. This reviewer was fully enthusiastic until the PK/PD experiments described in the next to last paragraph of the Results/Discussion, which were disappointing. The manuscript somewhat abruptly ends with these results and there is little discussion of what may be causing the poor bioavailability and how it might be improved, and why GRL0617 and derivatives such as XR8-23 and XR8-24 (Shen et al., J. Med. Chem 2021) seem to exhibit superior PK/PD profiles. This reviewer suggests the following points to consider as ways of improving the manuscript:

*The manuscript could be edited to appeal to a broader audience. Showing a simple chemical drawing of the key molecules focused on in the study earlier in the manuscript and an overview of how they form covalent adducts with PLpro might be helpful. A schematic summarizing the sites where modifications to GRL0617 were designed along with the rationale for such modifications might be helpful to nonspecialists.

*The manuscript could benefit from a standalone discussion section describing how the compounds described in and major conclusions of this study fit into the broader body of work focused on PLpro inhibition. As noted above, a discussion of why compound 7 might exhibit such poor a PK/PD profile along with ideas for how it might be improved, as well as a discussion of why related noncovalent GRL0617 analogs exhibit superior PK/PD profile would be beneficial. How is this study novel and how does it help push the field forward?

*Along these lines, did the authors test the PK/PD profile of compounds 5/6 or 14/15?

*For the PLpro-compound 7 crystal structure, composite omit map electron density should be shown for compound 7 shown in a way that the covalent linkage and other atoms of the compound can be observed.

Typos and other minor considerations:

*Please define units in Fig. S4

*Figure S6- I cannot tell which of the initial rate graphs corresponds to which compound.

*Figure S7 compares compounds 11 and 13, not 11 and 12 as indicated in the text on page 6.

*The text on page 11 refers to Supplementary Figure S11. I believe this is meant to Supplementary Figure S10 as there seems to be no figure S11.

Reviewer #2 (Remarks to the Author):

This manuscript by Sanders and co-workers describes the design and characterization of covalent inhibitors of the SARS-CoV-2 papain-like protease (PLpro). PLpro is required for processing the nonstructural proteins; it also regulates the host immune response by cleaving ubiquitin and ubiquitin-like protein such as ISG15. Here, the authors evaluated a series of acetylaceto-hydrazine-based covalent PLpro inhibitors and identified molecules with low nanomolar potency in both enzymatic and cellular assays. The author also provided structural information verifying the covalent inhibition. Too many papers have been published on developing highly druggable 3CLpro inhibitors, culminating with the FDA approval of Paxlovid; very few successful efforts have been reported on PLpro inhibition as it is difficult to drug with featureless and flexible binding pockets. The manuscript is clearly written, the experimental design is elegant, and the results make a significant contribution to the field as the first

non-peptide-based covalent inhibitors with low nanomolar potency.

Comments:

1. In Figure 4b, the authors plotted the Kobs against cpd 7 concentrations to derive Kinact and Ki. Apparently, the curve reaches its inflection point at 0.3 uM. I will suggest running a couple of more concentrations as the authors did in Figure S6 and fitting the data using the Michaelis–Menten equation to acquire both Kinact and Ki ($K_{obs} = \frac{K_{inact}[I]}{K_i + [I]}$). The second-order rate constant (k_{inact}/K_i) is only valid when the testing concentrations are significantly below K_i (at least 10-fold).
2. It will help the readers to label the electrophilic carbon that reacts with cysteine in Figure 3. Some compounds are attacked at alpha carbon (7-10), while others are attacked at beta carbon (11-13). The general trend suggests that the alpha carbon is at the right distance and geometry to react with Cys111. The author can add this to the discussion. It will be very interesting to make and test a succinate control compound to dissect further the contribution of binding and reactivity attributed to the potency of compound 7.
3. The section for the analysis of the co-crystal structure is underdeveloped. The authors need to overlay with more co-crystal structures to highlight the movement of key residues. There is a key Leu162, that is a gatekeeper for accessing the active site channel. In the GRL conformation, it's folded to lock the entrance, while your structure is likely propped up via the hydrogen bond network with your acetohyrazine linker.
4. It's pretty intriguing the authors chose to use GSH as a reducing agent in the enzymatic assays against the less nucleophilic TCEP.
5. The headline for Table 3 is incorrect? Line 288 indicated that it's a virus yield reduction assay.

We note that Reviewer 1 stated of our original submission that the “*idea of using GRL0617 as a scaffold for design of a covalent PLpro inhibitor is clever and the study appears to be extremely rigorous.*” Similarly, Reviewer 2 found that, “*The manuscript is clearly written, the experimental design is elegant, and the results make a significant contribution to the field as the first non-peptide-based covalent inhibitors with low nanomolar potency.*” However, both Reviewers raised a few points that required our attention. Their comments have certainly led to improvements in the revised manuscript. Below, we reproduce the Reviewer comments (in italics) and provide detailed responses to each point. Where relevant, revised text is shown in blue.

REVIEWER COMMENTS

Reviewer #1 (Remarks to the Author):

In this manuscript, Sanders et al. describe GRL0617-based covalent inhibitors of the SARS-CoV-2 papain-like protease. This involved design and synthesis of a number of different analogs followed by extensive characterization of these molecules in a comprehensive combination of in vitro, cell-based, and animal studies. Overall, idea of using GRL0617 as a scaffold for design of a covalent PLpro inhibitor is clever and the study appears to be extremely rigorous. This reviewer was fully enthusiastic until the PK/PD experiments described in the next to last paragraph of the Results/Discussion, which were disappointing. The manuscript somewhat abruptly ends with these results and there is little discussion of what may be causing the poor bioavailability and how it might be improved, and why GRL0617 and derivatives such as XR8-23 and XR8-24 (Shen et al., J. Med. Chem 2021) seem to exhibit superior PK/PD profiles. This reviewer suggests the following points to consider as ways of improving the manuscript:

**The manuscript could be edited to appeal to a broader audience. Showing a simple chemical drawing of the key molecules focused on in the study earlier in the manuscript and an overview of how they form covalent adducts with PLpro might be helpful. A schematic summarizing the sites where modifications to GRL0617 were designed along with the rationale for such modifications might be helpful to nonspecialists.*

Response: To improve readability for a broader audience we have added the following new figure (Figure 2), which clearly shows our design strategy and have modified the Introduction accordingly.

Figure 2. Design strategy for covalent PLpro inhibition. (a) X-ray co-crystal structure of ubiquitin-propargylamine (cyan) covalently bound to Cys111 in PLpro (tan) from PDB entry 6xaa.¹² Selected residues from PLpro and the LRGG motif of ubiquitin (cyan) are labeled and shown in stick representation. (b) Crystal structure of GRL0617 (cyan) bound to PLpro from PDB entry 7cmd.¹⁸ The distance between S_γ of Cys111 and the tolyl methyl of GRL0617 is labeled. (c) Components of covalent PLpro inhibitor candidates consisting of various electrophiles, a Gly-Gly mimetic linker, and the GRL0617 core. Reactive carbons on electrophiles are labeled with asterisks. (d) Mechanism of covalent bond formation between Cys111 and an inhibitor candidate with a fumarate ester electrophile.

**The manuscript could benefit from a standalone discussion section describing how the compounds described in and major conclusions of this study fit into the broader body of work focused on PLpro inhibition. As noted above, a discussion of why compound 7 might exhibit such poor a PK/PD profile along with ideas for how it might be improved, as well as a discussion of why related noncovalent GRL0617 analogs exhibit superior PK/PD profile would be beneficial. How is this study novel and how does it help push the field forward?*

Response: We agree that the manuscript ended abruptly and lacked sufficient discussion of the poor bioavailability, possible ways to improve it, and how other groups have developed inhibitors with superior PK/PD profiles. We have modified the text to include a comprehensive, standalone Discussion section in which we address the Reviewer's points. Specifically, we discuss the novelty of our first-in-class covalent PLpro inhibitors that display improved efficacy over noncovalent analogs. We discuss GRL0617 and the recently reported 2-phenylthiophene analogs that display much-improved PK properties. We then propose specific chemical modifications that could improve the metabolic liabilities and poor PK properties of compound 7. We also discuss alternative electrophiles that could be pursued in future efforts and finish the manuscript with a few final thoughts on covalent inhibition.

**Along these lines, did the authors test the PK/PD profile of compounds 5/6 or 14/15?*

Response: Only compounds that showed sufficient antiviral activity were considered for follow-up PK characterization. Compound 5 is only a 24- μ M inhibitor of PLpro. Given that there is

typically a loss of potency when moving from target to whole virus assays, this compound did not meet our criteria for progression to CPE assays. Compound **6** was not tested. For **14** and **15**, the IC₅₀s were 100 uM and 6.2 uM, respectively, but neither compound was protective in CPE assays with Vero E6 cells. Given that none of these exhibited an improved biological profile compared with **7** and that these compounds share the same base structure, it is highly unlikely that their ADME profiles would be improved. Thus, compounds 5/6 and 14/15 did not meet our criteria for progression to PK studies. We note here that the Reviewer did not specifically request these experiments. However, the Reviewer did also note that there are other reported compounds that exhibit in vivo exposure so we have added some discussion around these compounds (XR8-23 and XR8-24) to the manuscript. We did test **9** and **14** for metabolic stability. We have elaborated on these results in the revised manuscript.

**For the PLpro-compound 7 crystal structure, composite omit map electron density should be shown for compound 7 shown in a way that the covalent linkage and other atoms of the compound can be observed.*

Response: **Figure 7c** now shows a composite omit map ($\sigma = 1.0$), which clearly shows the electron density for the covalent bond between Cys111 and compound **7**.

Typos and other minor considerations:

**Please define units in Fig. S4*

Response: The units correspond to those in the x and y axes of the plot, i.e., uM/s for v_{max} and uM for K_m. We have added a sentence to the Figure caption for clarification.

**Figure S6- I cannot tell which of the initial rate graphs corresponds to which compound.*

Response: We have relabeled **Figure S6** to make it more intuitive for readers. Data in the left column (**Ia-e**) are initial rates determined at various inhibitor concentrations, and data in the right column (**IIa-e**) are the k_{obs} values determined from **Ia-e** \pm 95% confidence interval of the nonlinear regression.

**Figure S7 compares compounds 11 and 13, not 11 and 12 as indicated in the text on page 6.*

Response: Both **Figures S6** and **S7** are now referenced in this sentence on page 6 as all three compounds are compared in this sentence.

**The text on page 11 refers to Supplementary Figure S11. I believe this is meant to Supplementary Figure S10 as there seems to be no figure S11.*

Response: We have corrected this error.

Reviewer #2 (Remarks to the Author):

This manuscript by Sanders and co-workers describes the design and characterization of covalent inhibitors of the SARS-CoV-2 papain-like protease (PLpro). PLpro is required for

processing the nonstructural proteins; it also regulates the host immune response by cleaving ubiquitin and ubiquitin-like protein such as ISG15. Here, the authors evaluated a series of acetylacetohydrazine-based covalent PLpro inhibitors and identified molecules with low nanomolar potency in both enzymatic and cellular assays. The author also provided structural information verifying the covalent inhibition. Too many papers have been published on developing highly druggable 3CLpro inhibitors, culminating with the FDA approval of Paxlovid; very few successful efforts have been reported on PLpro inhibition as it is difficult to drug with featureless and flexible binding pockets. The manuscript is clearly written, the experimental design is elegant, and the results make a significant contribution to the field as the first non-peptide-based covalent inhibitors with low nanomolar potency.

Comments:

1. In Figure 4b, the authors plotted the Kobs against cpd 7 concentrations to derive Kinact and Ki. Apparently, the curve reaches its inflection point at 0.3 uM. I will suggest running a couple of more concentrations as the authors did in Figure S6 and fitting the data using the Michaelis–Menten equation to acquire both Kinact and Ki ($K_{obs} = \frac{K_{inact}[I]}{K_i + [I]}$). The second-order rate constant (k_{inact}/K_i) is only valid when the testing concentrations are significantly below K_i (at least 10-fold).

Response: We attempted experiments like those shown in Figure S6 a and b using both higher and lower inhibitor concentrations. At higher inhibitor concentrations the rate of covalent adduct formation was too fast to obtain a linear fit with sufficient points. At lower inhibitor concentrations longer preincubation times are required and enzyme stability becomes limiting. More sophisticated equipment would be required for accurate measurement of K_{obs} at higher inhibitor concentrations to catch the plateau. However, we have added a sentence in the Methods section pointing out that the k_{inact}/K_i measurements may be inaccurate when this condition is not met:

We note that k_{inact}/K_i is only valid when the testing concentrations are at least 10-fold below K_i , so there may be inaccuracies when this condition is not met.

2. It will help the readers to label the electrophilic carbon that reacts with cysteine in Figure 3. Some compounds are attacked at alpha carbon (7-10), while others are attacked at beta carbon (11-13). The general trend suggests that the alpha carbon is at the right distance and geometry to react with Cys111. The author can add this to the discussion.

Response: We have labeled the electrophilic carbons of each candidate warhead in the newly added Figure 2. We have also added the following statement to the Discussion:

Whereas some of our candidate inhibitors are attacked at the α carbon (7-10), others are attacked at the β carbon (11-13). The general trend from in vitro inhibition assays suggests that the α carbon is at the right distance and geometry to react with Cys111.

It will be very interesting to make and test a succinate control compound to dissect further the contribution of binding and reactivity attributed to the potency of compound 7.

Response: As part of a separate project, we did synthesize the amino analog of compound **7** with the fumarate reduced to succinate. This compound did not inhibit PLpro (IC50 >50 uM).

3. The section for the analysis of the co-crystal structure is underdeveloped. The authors need to overlay with more co-crystal structures to highlight the movement of key residues. There is a key Leu162, that is a gatekeeper for accessing the active site channel. In the GRL conformation, it's folded to lock the entrance, while your structure is likely propped up via the hydrogen bond network with your acetohyrazine linker.

Response: **Figure 7d** now shows superpositions of the co-crystal structure of PLpro and compound **7** with several other crystal structures. A more detailed version showing additional structures has also been added to the Supplementary Material (**Figure S11**). The relevant text from the figure caption is:

Superposition of selected structures highlighting the positions of the side chain of Leu162 (sticks) and the BL2 loop (cartoon) in the absence and presence of selected inhibitors: Ligand-free (PDB entry 6W9C, light green), glycerol-bound (PDB entry 6WZU, purple), GRL0617-bound (PDB entry 7CMD, light purple), and compound **7**-bound (this work; cyan). Additional structures are shown in **Figure S11**.

4. It's pretty intriguing the authors chose to use GSH as a reducing agent in the enzymatic assays against the less nucleophilic TCEP.

Response: In the cited references, DTT or GSH were used. GSH was used in this study due to availability in the laboratory and the choice of reducing agent was not explored further. No GSH adducts were observed in the MS studies.

5. The headline for Table 3 is incorrect? Line 288 indicated that it's a virus yield reduction assay.

Response: The Reviewer is correct. We have revised the table header to reflect that the data are from a virus yield reduction assay.

REVIEWERS' COMMENTS

Reviewer #1 (Remarks to the Author):

In this revised manuscript, Sanders et al. have adequately addressed all of the reviewer concerns. The revised manuscript is more accessible to the broad readership of Nature Communications and the new Discussion section highlights the significance of this study in the broader context of the field. The more thorough analysis of the co-crystal structure also adds to the impact of the study. This reviewer supports publication of this manuscript.

Reviewer #2 (Remarks to the Author):

I'm happy with the changes made by the authors. The newly made figures are much more clear to readers. I thank them for their efforts to address my comments and for summarizing them clearly in their response letter. I want to emphasize again that this is a significant study on PLpro, and I recommend publishing the paper.